

# Enhancing geotechnical damage detection with deep learning: a convolutional neural network approach

Thabatta Moreira Alves de Araujo[1,2], Carlos André de Mattos Teixeira[1] and Carlos Renato Lisboa Francês[1]

[1] High Performance Network Planning Laboratory, Federal University of Pará, Belém, Pará, Brazil
[2] Departament of Computing, Federal Center for Technological Education of Minas Gerais, Divinópolis, Minas Gerais, Brazil

## ABSTRACT

Most natural disasters result from geodynamic events such as landslides and slope collapse. These failures cause catastrophes that directly impact the environment and cause financial and human losses. Visual inspection is the primary method for detecting failures in geotechnical structures, but on-site visits can be risky due to unstable soil. In addition, the body design and hostile and remote installation conditions make monitoring these structures inviable. When a fast and secure evaluation is required, analysis by computational methods becomes feasible. In this study, a convolutional neural network (CNN) approach to computer vision is applied to identify defects in the surface of geotechnical structures aided by unmanned aerial vehicle (UAV) and mobile devices, aiming to reduce the reliance on human-led on-site inspections. However, studies in computer vision algorithms still need to be explored in this field due to particularities of geotechnical engineering, such as limited public datasets and redundant images. Thus, this study obtained images of surface failure indicators from slopes near a Brazilian national road, assisted by UAV and mobile devices. We then proposed a custom CNN and low complexity model architecture to build a binary classifier image-aided to detect faults in geotechnical surfaces. The model achieved a satisfactory average accuracy rate of 94.26%. An AUC metric score of 0.99 from the receiver operator characteristic (ROC) curve and matrix confusion with a testing dataset show satisfactory results. The results suggest that the capability of the model to distinguish between the classes 'damage' and 'intact' is excellent. It enables the identification of failure indicators. Early failure indicator detection on the surface of slopes can facilitate proper maintenance and alarms and prevent disasters, as the integrity of the soil directly affects the structures built around and above it.

Corresponding author
Thabatta Moreira Alves de Araujo, thabattaa@gmail.com

## INTRODUCTION

From a geotechnical perspective, most natural catastrophes stem from diverse geodynamic events, including landslides, slope instability, and other phenomena. These natural disasters and others can have wide-ranging impacts, affecting the environment, causing financial

and human losses, and directly impacting the ecosystem (*Solórzano et al., 2022*). In environments where structural stability is paramount, it is imperative to remain watchful for any indications of failure on the surface. These indicators may be early warning signs of potential structural compromise that could lead to catastrophic failure. By being mindful of these warning signals, we can take proactive measures to prevent such failures. However, predicting failures and preventing geotechnical disasters is a challenging task (*Handy, 2007*) and multi-approach (*Lima Jr., Venturini & Benallal, 2010*).

Detecting damage on the surface provides essential information about the behaviour of the soil and rocks of geotechnical structures such as excavations, dams, embankments, containment works, and natural geological formations. A primary approach to identifying failures is through visual inspection during site visits, which play a crucial role in geotechnical engineering and engineering geology, providing essential information about the performance of the structure, (*Handy, 2007*; *Lim et al., 2021*; *Han et al., 2022*). Signs of damage, such as erosion, landslides, without ground cover, seepage, leakage, settlement, and shear displacement, are often visible on the surface and can be identified by the human eye. Although efficient, it is strongly dependent on the work of experts, *Pan & Chen (2015)*.

To aid on-site inspections, images, audio, and video records, resulting in relevant information for analysis regarding the structural safety of embankments, earth-rock dams, and slopes. Recently, technologies such as digital images, satellites, unmanned aerial vehicles (UAV), robots, computer-embedded boards, and computational intelligence for damage detection have emerged (*Greenwood et al., 2019*; *Kanellakis & Nikolakopoulos, 2017*). Computer vision enables remote image-based sensing to aid visual inspections (*Jang, Kim & An, 2019*; *Lei et al., 2020*; *Tian et al., 2023*; *Cha, Choi & Büyüköztürk, 2017*; *Di et al., 2014*). In this way, visual analysis of apparent damage is crucial for identifying potential failures (*Kanellakis & Nikolakopoulos, 2017*). However, recent research suggests that the use of advanced technologies such as machine learning, deep learning, computer vision, and big data in geotechnical engineering, particularly in the context of soil and rock structures, remains relatively uncommon (*Zhang, Li & Li, 2021*; *Phoon & Zhang, 2023*). In practical terms, the proportion of research utilising these technologies in soil and rocks is comparatively lower in this field than in other applications (*Zhang, Li & Li, 2021*). In specific computer vision applications, most research focuses on geotechnical structures in concrete (*Li et al., 2019b*; *Lei et al., 2020*). Therefore, the opportunities for research and application of classic and advanced models in soil and rock structures are vast (*Zhong et al., 2020*; *Zhang, Li & Li, 2021*).

The convolutional neural network (CNN) for computer vision is extensively applied in image pattern recognition (*Li et al., 2022*), and its architectures have already garnered prominence in addressing computer vision challenges. These networks are equipped with layers capable of automatically extracting features from images during the training phase. This feature includes edges, textures, shapes, patterns, and other visual components pertinent to the task. Support vector machine (SVM) and random forest, among other traditional techniques, stand as robust classification algorithms, but features are extracted manually (*Myagila & Kilavo, 2022*; *Sudha et al., 2019*). The circumstances could hinder the practical execution of this. Thus, these methodologies do not fall within the purview

of this study, as our aim resides in exploring CNN architectures within an area that remains relatively uncharted, such as geotechnical. The CNN approach based on image analysis and classification presents a viable solution that enables the merging of monitoring systems aided by drones, satellites, and other image-generating systems (*Limão, de Araújo & Frances, 2023*; *Han et al., 2022*). This approach could enhance the monitoring system, improving accuracy, objectivity, and processing speed.

So, we introduce utilising a CNN architecture for building a binary classifier capable of identifying surface damage in geotechnical structures through remotely acquired images by UAV. This constitutes a computer vision conundrum, where the resolution is achieved by categorising the images into two distinct classes based on the likelihood of image deterioration. A previous study conducted in *Limão, de Araújo & Frances (2023)* uses generic images from the Internet in a similar approach to ours. However, the manual building dataset required significant effort to screen images that had a license for recreational or educational use. Moreover, there were difficulties in finding images that adequately represent damage to geotechnical surfaces. Our current proposal uses an authentic dataset (*De Araujo, 2023*). It is worth noting that the volume of images collected by UAVs and mobile devices for inspection is significant but considerably redundant for slopes. The uniform and typical vegetation coverage under healthy conditions reduces the diversity of the dataset, which can lead to the model being overfitted. When it comes to soil with visible surface damage, it is quite challenging to obtain a set of images that can help build a dataset with several categories or classes of damage. Obtaining such a dataset can prove challenging for failures or catastrophes, as these occurrences are infrequent, which also limits the dataset. In this way, we decided to go with a binary classifier.

This article delves into various CNN architectures and explores how to strike a balance among them when designing a model for a real-world application. However, to obtain a suitable architecture for classifying images, we need to consider the size of our dataset and the computational resources available to perform the model. We chose to investigate a leaner architecture, reducing the complexity of the model, to promote computational efficiency without sacrificing accuracy (*Brigato & Iocchi, 2021*; *Foroughi, Chen & Wang, 2021*). Although there are robust and complex CNN architectures available for computer vision applications, such as Visual Geometry Group (VGG), Inception and ResNet (*Yadav & Jadhav, 2019*; *Abedalla et al., 2021*) with denser layers, they are often more challenging to implement because they require greater computing resources and abstract representations as the network becomes deeper (*Brigato & Iocchi, 2021*; *Darapaneni, Krishnamurthy & Paduri, 2020*).

To address all the aforementioned issues, this article proposes the construction of a dataset comprising authentic images and a 3-layer CNN with two fully connected layers that can detect apparent damage based on the visual diagnosis from images. We utilise techniques to prevent overfitting, including regularisation methods such as dropout and data augmentation, as well as Adam optimiser. The database images contain visual indicators of landslides, without ground cover and superficial erosion images on slopes. The slopes belong to geotechnical structures such as excavations, dams, embankments, containment works, and some natural geological formations. Mapping landslides, without

ground cover and erosion hazards, contributes to solving problems of soils (*Terzaghi, Peck & Mesri, 1996*). As a result, image analysis and anomaly detection through CNN enable early damage classification, which can prevent accidents and maintain soil and rock integrity and the surrounding environment. This also mitigates disasters and geotechnical damage in challenging and inaccessible hostile environments.

To summarise, this article presents the following contributions: (1) We have collected, labelled, prepared and made publicly available representative images that depict visible damage on the surface of slopes; (2) We have conducted experiments with low-complexity CNN architectures; (3) We have demonstrated that low-complexity CNNs, combined with regularisation techniques, optimisation of learning, and data augmentation, achieve satisfactory performance for redundant and smaller datasets; (4) We have developed a model for binary classification of images with satisfactory performance, requiring little computational resources for processing (single CPU); and (5) We have explored a strategy that leverages data acquisition by UAV and mobile devices for remote inspection and detection of damage visible to the naked eye on the surface of geotechnical structures by way of CNN.

'Related Work and Background Concepts' presents the related work and the concepts of the leading technologies, along with their pros and cons and current state. 'Overview of the Proposed Method' provides an overview of the methodology. 'Building Classifier for Detecting Surface Damage on Geotechnical Structures' presents the proposed classifier to detect surface damage in geotechnical structures. The section includes performance discussions and analysis of the developed CNN architecture, emphasising the training and testing details that yielded the most accurate results in detecting geotechnical damage. 'Conclusion' concludes the paper by analysing the results and offering suggestions to enhance the proposed approach.

## RELATED WORK AND BACKGROUND CONCEPTS

This section will briefly review the literature, including various research studies and proposals on computer vision applied to damage prediction methods. In the development of this work, bibliographic research was conducted, focusing on well-established articles that would serve as the foundation for new artificial neural network (ANN) implementations.

This article aims to integrate the main themes and technologies discussed in each reviewed article to propose an architecture that would achieve optimal performance for the neural network. Table 1 highlights the computer vision-based CNN for damage detection among the proposed proposals studied.

Unlike the approach proposed in this paper, similar solutions based on CNNs combine various computer vision methods. These solutions demonstrate the application of multiple convolutional layers to effectively process the available dataset and extract relevant information from the image patterns (*Lindsay, 2021*; *Tan, Guo & Poh, 2021*; *Bari et al., 2021*). Examples are applications based on computer vision medicine (*Yadav & Jadhav, 2019*; *Cano et al., 2021*), agriculture (*Kattenborn et al., 2021*; *Lu, Tan & Jiang, 2021*), remote sensing (*Li et al., 2018*), manufacturing (*Affonso et al., 2017*) and easy recognition (*Ekundayo & Viriri, 2021*).

**Table 1  Used base-related works.**

| Studied research | Research summary |
| --- | --- |
| *Li et al. (2019a)* | Application scenario for employing YOLOv3 to crack detection in floodgate dam surface and share its effects. |
| *Lei et al. (2020)* | Vision-based concrete crack detection method that includes images collected by unmanned aerial vehicles, pre-processing algorithm, crack central point method, and the support vector machine model–based classifier. |
| *Cha, Choi & Büyüköztürk (2017)* | Vision-based method using a deep architecture of convolutional neural networks (CNNs) for detecting concrete cracks without calculating the defect features. |
| *Tian et al. (2023)* | A neural network is trained to detect falling rocks from captured images to extract motion information and reconstruct impact force. Additionally, a two-dimensional distribution map of velocity amplitude is generated to track the spatial deflection pattern of the flexible barrier system during rockfall impact. |
| *Han et al. (2022)* | Deep learning-based clayey soil crack detection, localisation and segmentation. |
| *Lim et al. (2021)* | A faster region-based CNN is constructed and applied to the combined vision and thermographic images for automated detection and classification of surface and subsurface corrosion in steel bridges. |
| *Zhang & Tong (2023)* | Proposes a water level measurement method based on computer vision, constructs a high-resolution representation deep learning regression model for water line positioning in the pre-processed image |
| *Li et al. (2018)* | A comparative analysis regarding the performances of typical DL-based. Remote Sensing image classification. |
| *Jang, Kim & An (2019)* | Proposes a deep CNN model for automatic concrete crack detection, which utilises hybrid images combining vision and infrared thermography. This approach improves the detectability of cracks. |
| *Di et al. (2014)* | A machine learning approach to crater detection from image and topographic data. First, detecting square regions which contain one crater with the use of a boosting algorithm and second delineating the rims of the crater in each square region. |
| *Shi et al. (2016)* | A method for detecting and classifying underwater dam cracks based on visual imagery and fuzzy evidence combination. |
| *Ramandi et al. (2022)* | An algorithm for automatic fracture detection based on grayscale 3D CT images is available. The first step is a low-level early vision stage, which identifies potential fractures, and a high-level interpretative stage, to extract planar fractures from digital rock images. |

Moreover, machine learning approaches in pattern recognition are explored in *Figueiredo et al. (2011)*, *Shahin, Maier & Jaksa (2003)*, *Dang et al. (2022)*, *Pan & Chen (2015)*, *Salajegheh, Mahdavi-Meymand & Zounemat-Kermani (2018)*, *Azimi & Pekcan (2020)* and *Claro et al. (2020)*; *Jung, Berges & Garrett (2014)*, while important concepts about

computer vision are explained in *Murray & Perronnin (2014)*; *Basha et al. (2020)*; *Girshick (2015)*; *Szegedy et al. (2015)*.

Finally, studies have been conducted using customised, low-complexity models for image classification in computer vision tasks without data augmentation. We want to highlight a study by *Brigato & Iocchi (2021)* examining the complexity of multiclass classifiers. This study compares state-of-the-art networks (including popular deep learning models, such as ResNet) and proposes simpler alternatives for small datasets in image classification tasks. The results present in the study suggest that relatively simple networks can play a significant role in being less prone to overfitting and generalizing better when facing small datasets. Furthermore, *Foroughi, Chen & Wang (2021)* propose a binary classifier using a custom CNN on a small dataset, and with the help of the data augmentation technique, it reaches a dataset of 534 images. A comparative analysis is carried out with the VGG16 model to validate the model. The metrics indicate superior custom and low complexity model performance compared to VGG.

## Surface damage

Geotechnics is an area of civil engineering that investigates the mechanics of rocks and soils, geological engineering, and similar related fields (*Handy, 2007*; *Terzaghi, Peck & Mesri, 1996*). Therefore, the following terms are defined to improve the comprehension of this study.

- Rock: A hard or firm mass that was intact and in its natural place prior to the start of movement (*Varnes, 1978*).
- Soil: An aggregate of solid particles, typically minerals and rocks, that were transported or formed by rock weathering in place. Gases or liquids that fill the pores of the soil become part of the soil (*Varnes, 1978*).
- Earth: Material in which 80% or more of the particles are smaller than 2 mm, the upper limit of sand-sized particles (*Varnes, 1978*).

Geotechnical damage detection usually requires a rigorous technical evaluation assisted by specialised procedures and tools (*Das, 2017*; *Varnes, 1978*; *Lima Jr., Venturini & Benallal, 2010*). Usually, damage detection technology is based on on-site inspections, sensors, and laboratory tests. Visual inspections are the most explored (*Han et al., 2022*). The identification of a single exact cause of failure is often unattainable. A combination of geology, topographic, climatic, human, and other elements usually contributes to the triggering of damage (*Volkwein et al., 2011*; *Khan et al., 2021*).

Observation of the surface of the structure enables the identification of failures, including those without vegetation cover, erosion, and mass movements. Visual identification is a vital component of soil analysis without instruments or laboratory facilities, facilitating methodical description (*Das, 2011*). Therefore, the following are two types of common geotechnical structural failures that are quickly apparent to the naked human eye during inspections.

### Superficial erosion

Erosion occurs due to the wear of soil particles on the surface, entrapment of rainwater, wind, temperature fluctuations, and other geological agents, including gravitational forces (*Borrelli et al., 2021*). Soil erosion is a significant indicator of failure, which occurs when soil resistance drops due to various factors, including reduction of matrix suction stress, discontinuities, modification of the structure of sensitive soils, liquefaction of saturated sand, and loss of cohesion.

The erosive processes of the slopes are controlled by natural and anthropic factors, including rain erosivity, soil erodibility (vulnerability), the nature of the vegetation cover, vegetation cover, slope characteristics and types of soil use and occupation (*Hudson, 1961*).

Irregular terrain is more vulnerable to water erosion since splashing, surface runoff, and transportation all have more pronounced effects on steep slopes. Furthermore, soil characteristics and properties impact its vulnerability by affecting factors such as water infiltration rate, permeability, and water absorption capacity, as well as indicating the potential for dispersion, splash, abrasion, and transport caused by rainfall and flooding (*Nguyen & Indraratna, 2020*; *Cooke & Doornkamp, 1990*).

Vegetation cover reduces soil erosion rates because it protects against the impact of rain, decreases the amount of water available for surface runoff, decreases surface runoff speed, and increases the capacity of the soil for water infiltration (*Cooke & Doornkamp, 1990*). Therefore, vegetation cover is an important aspect when analysing vulnerable soil and its lack indicates possible failure.

### Landslide

The term 'landslide' pertains to mass movements characterised by a distinct zone of weakness, separating the sliding material from the more stable underlying strata, as depicted in Fig. 1. This geological phenomenon is intricate, influenced by a multitude of factors, including slope geometry, soil quality, moisture content, precipitation, vegetation index, construction activities in the region, surface load, and proximity to roads and rivers (*Terzaghi, Peck & Mesri, 1996*; *Aziz et al., 2021*). The influence of these factors may vary between regions and geological and topographic conditions (*Khan et al., 2021*).

Landslides are classified by type of material and type of movement. Usually, the material is rock or soil. The two main types of slides are rotational slides and translational slides. The landslide phenomena can be a combination of two or more of the movement types (*Varnes, 1978*).

This article aims to investigate the application of computer vision, in combination with high-performance processing techniques such as CNN, to detect damage in geotechnical structures. The reliability of such structures is crucial for their effective functioning and the safety of the surrounding areas.

The following subtopics will provide the definitions and concepts necessary to understand the proposed solution effectively. These subtopics are arranged in the following order: deep learning for image classification, convolutional layer, activation layer ReLu, pooling layer, dropout, Adam optimiser and model performance assessment.

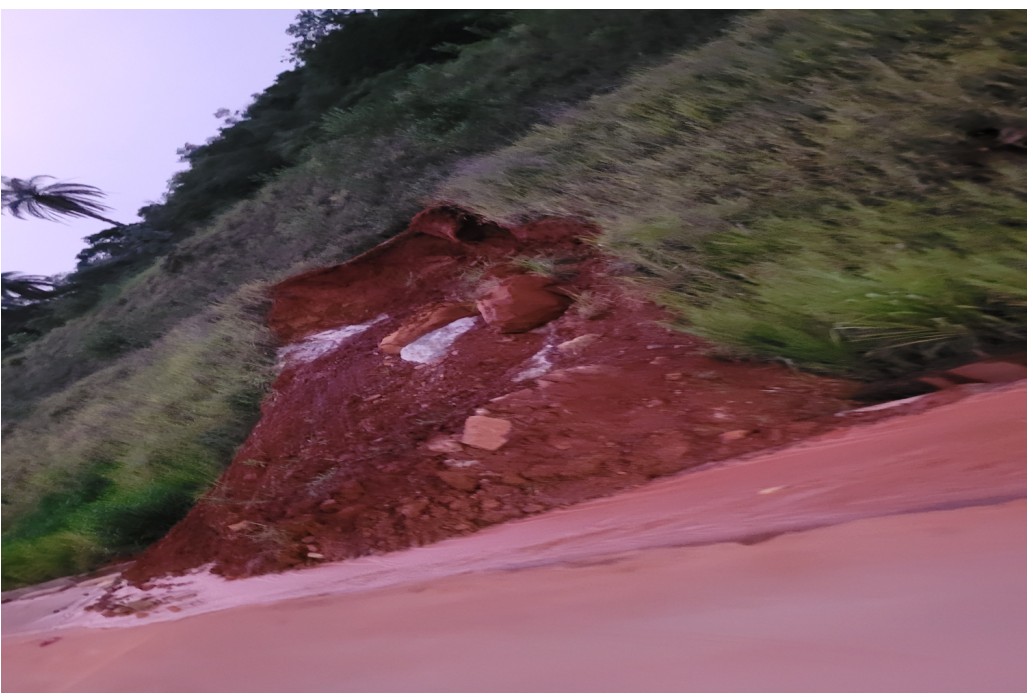

**Figure 1** **Landslide near Highway 381, Brazil 2022.**

## Deep learning for image classification

Deep learning (DL) is a concept that derives from the conventional neural network but outperforms it by employing layers that operate on the network with the input data. Therefore, it is considered a deep network because multiple layers can contain multiple operations. It outperforms other types of machine learning (ML) architectures for processing unstructured data formats such as video and images (*Alzubaidi et al., 2021*; *Limão, de Araújo & Frances, 2023*; *Cha, Choi & Büyüköztürk, 2017*; *Lei et al., 2020*; *Jang, Kim & An, 2019*; *Lindsay, 2021*; *Tan, Guo & Poh, 2021*; *Bari et al., 2021*). Among all types of deep neural networks, the model stands out: the CNN.

The CNN structure (Fig. 2) was inspired by the actual operation of vision itself, and it has become a successful tool in computer vision and state-of-the-art models of neural activity and visual tasks. They start their process by convolving a set of filters with the input and rectifying the output, leading to feature maps similar to the planes of S cells in neurorecognition (*Lindsay, 2021*). Figure 2 shows a representative CNN structure classifier containing two blocks formed by convolutional, max pooling and fully connected layers. CNN architectures are regarded as deep due to their successive incorporation of convolutional layers. These layers carry out filtering and pooling operations, thereby extracting features that serve as inputs to a conventional neural network (fully connected). As a result, simpler CNN may prove to be more computationally efficient, which is crucial in environments with constrained resources. Consequently, simpler CNNs may be more

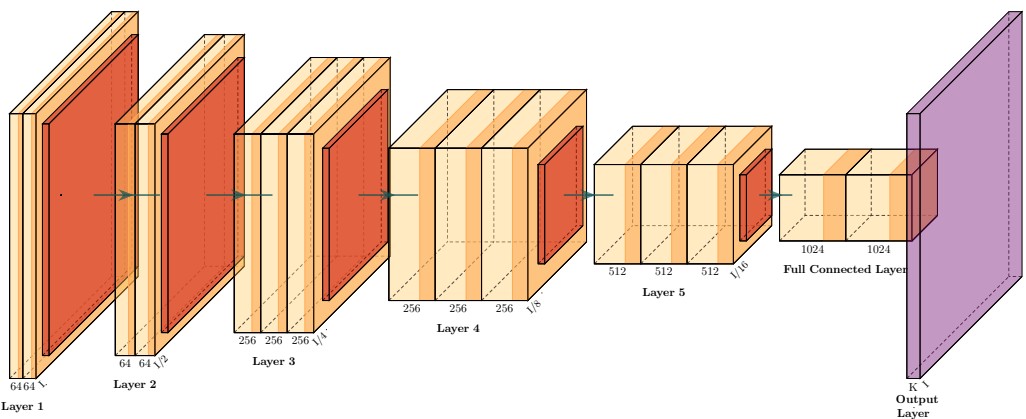

512  512  512  1/16
**Full Connected Layer**

1024  1024
**Layer 5**

256  256  256  1/8
**Layer 4**

64  64  1/2
**Layer 3**

6464
**Layer 1**

**Layer 2**

K
**Output Layer**

**Figure 2** Illustrative CNN structure with five convolutional-pooling layers and two fully connected, adapted from *Iqbal (2018)*.

computationally efficient, a critical aspect in environments with constrained resources (*Darapaneni, Krishnamurthy & Paduri, 2020*).

Although there are many advantages to using CNN for computer vision, there are also some challenges. Overfitting is one of the most common problems faced by CNN (*Garbin, Zhu & Marques, 2020*; *Shorten & Khoshgoftaar, 2019*; *Cha, Choi & Büyüköztürk, 2017*). This occurs when the model fits very well with the training data but struggles to generalise with new data. Very complex networks can cause this, depending on the dataset or the need for regularisation techniques such as dropout (*Garbin, Zhu & Marques, 2020*; *Theodoridis, 2020*), data augmentation (*Abedalla et al., 2021*; *Shorten & Khoshgoftaar, 2019*; *Ottoni, de Amorim & Novo, 2022*), or adaptive optimisation (*Zhang, 2019*; *Garbin, Zhu & Marques, 2020*; *Zhang, Li & Li, 2021*). Additionally, problems may arise with updating weights due to gradient reduction or increase, which can lead to training instability (*Adnan et al., 2022*; *Abedalla et al., 2021*; *Zhang, 2019*). Another important point is the correct adjustment of hyperparameters (*Darapaneni, Krishnamurthy & Paduri, 2020*; *Ottoni, de Amorim & Novo, 2022*), which can be a complex task and require experience or automated optimisation methods.

A CNN comprises three scale dimensions that have a significant impact on its models. These dimensions include depth number of layers in the network), width (the number of channels in a layer), and resolution (image resolution) (*Abedalla et al., 2021*). This approach allows for multiple experiments, facilitating several iterations and extensive testing, resulting in diverse models and reducing some problems, such as overfitting and changes in gradient.

The following text provides a summary of some popular deep-learning models that are widely used in computer vision. These models include AlexNet, VGG, Inception V2, ResNet, and DenseNet (*Darapaneni, Krishnamurthy & Paduri, 2020*; *Abedalla et al., 2021*; *Ekundayo & Viriri, 2021*; *Yadav & Jadhav, 2019*; *Jang, Kim & An, 2019*). The AlexNet is a deep learning architecture that comprises five convolutional layers and 3 fully connected

layers (*Abedalla et al., 2021*). It employs the ReLU activation function and other techniques, such as dropout and data augmentation, to mitigate overfitting. Additionally, it addresses the issue of gradient vanishing, which is a common problem in deep learning. VGG, or Visual Geometry Group, is another deep learning model consisting of 16 or 19 convolutional layers using 3 × 3 filters (*Ekundayo & Viriri, 2021*; *Yadav & Jadhav, 2019*). The model follows a uniform approach in the architecture, which can demand significant processing power from the GPU or CPU due to a large number of layers and parameters. The Inception V2, or GoogLeNet, is a deep learning model distinguished by the "Inception" architecture (*Jang, Kim & An, 2019*). This architecture employs filters of different sizes, pooling layers, and 1 × 1 convolutions before larger 3 × 3 and 5 × 5 convolutions to reduce dimensionality (*Yadav & Jadhav, 2019*). This approach alleviates the need to choose the filter size manually. A ResNet, or Residual Networks, is a deep learning model that addresses the issue of gradient vanishing using residual blocks to train deep networks (*Abedalla et al., 2021*). This model enables training networks with hundreds of layers, which was previously a challenging task (*Darapaneni, Krishnamurthy & Paduri, 2020*). Lastly, DenseNet is a deep learning model characterised by dense connectivity between layers. Each layer receives activation from all preceding layers, allowing each layer direct access to features learned in all preceding layers (*Darapaneni, Krishnamurthy & Paduri, 2020*). This approach differs from traditional methods and has shown impressive results in various computer vision tasks.

In theory, increasing the number of layers (deeper layers), as seen in architectures like VGG, Incept and ResNet, for example, or augmenting the channels and image resolution, should improve the performance of the network. However, such modifications may impact the number of parameters and complexity of the model, making the CNN impractical—especially with limited resources (*Brigato & Iocchi, 2021*; *Darapaneni, Krishnamurthy & Paduri, 2020*). In some settings, the state-of-the-art can be improved using low-complexity models (*Brigato & Iocchi, 2021*).

### *Convolution layer*

The convolution layer is responsible for convolving the image patches to extract universal and increasingly complex features. The convolutional layer uses filters, Fig. 3, which perform convolution operations when scanning the input. These operations are performed on the entire image by sliding kernels and finding the dot product between the filter and the input image parts. Consider an input image $I$ with dimensions of height $H$, width $W$, and $C$ colour channels (*e.g.*, RGB, where $C = 3$). Alongside this input, we have a set of K convolution filters, each with a height and width of ($h \times w$) (also known as kernel size) and $C = 3$ channels to match the number of channels of the input image. The convolution operation on a single CNN layer can be precisely defined in Eqs. (1) and (2).

$$Z(x, y) = \sum_{c=1}^{C} \sum_{i=-(Fw-1)}^{Fw-1} \sum_{j=-(Fh-1)}^{Fh-1} I(x+i, y+j, c) \cdot K(i, j, c) \tag{1}$$

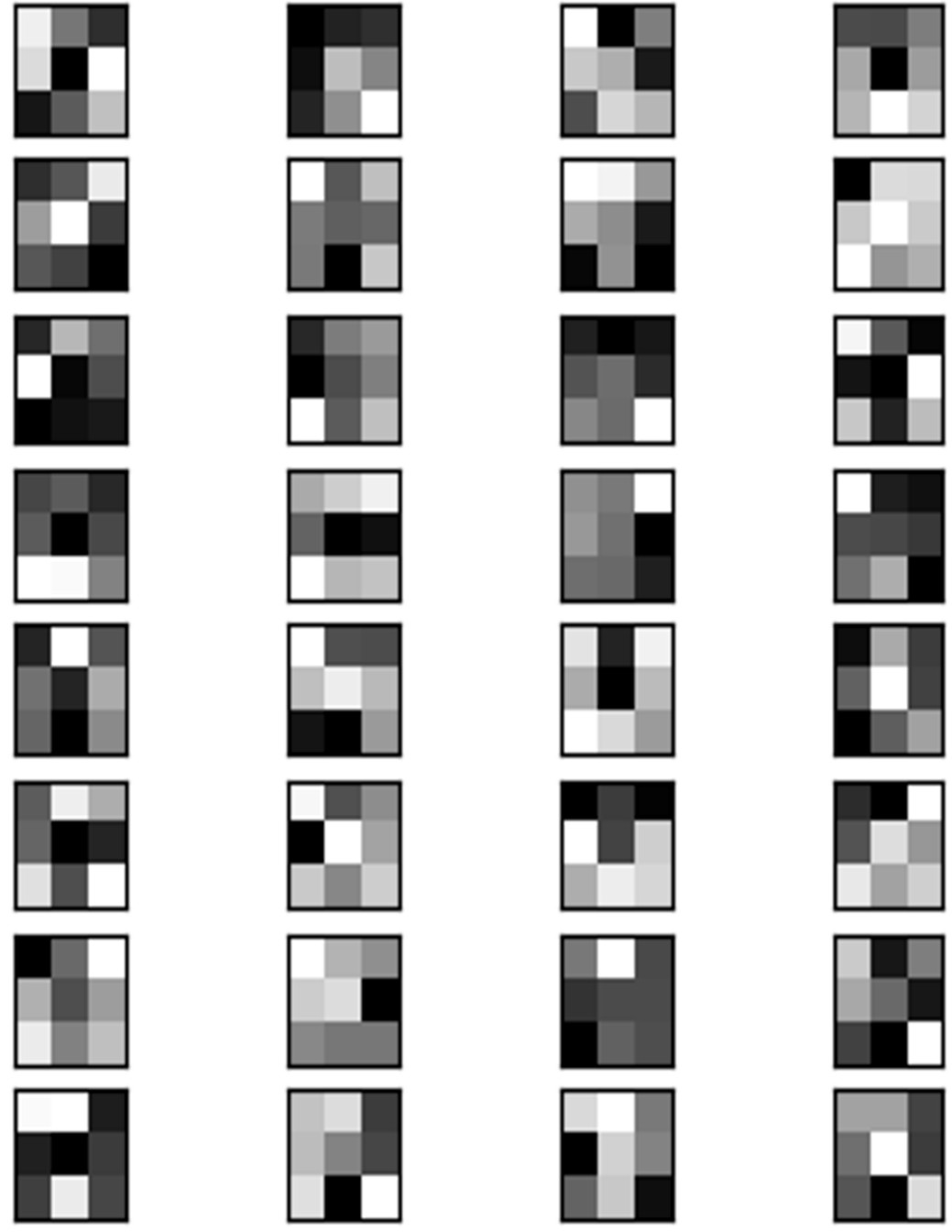

**Figure 3** Grayscale filters used in convolutional layer on image.

$$A(x,y) = f(Z(x,y)) \tag{2}$$

where $Z(x,y)$ is the output of the convolution operation at a specific location $(x,y)$ in the output feature map; $c$ is the channel index, representing one of the $C$ color channels in the input image and the corresponding filter channel; $i$ is the horizontal offset of the filter relative to the current location $(x,y)$ in the input image, and it ranges from $-((Fw-1)/2)$ to $((Fw-1)/2)$, where $Fw$ is the filter width; $j$ is the vertical offset of the filter relative to the current location $(x,y)$ in the input image, and it ranges from $-((Fh-1)/2)$ to $((Fh-1)/2)$, where $Fh$ is the filter height. $I(x+i,y+j,c)$ is the value of the input pixel at position $(x+i,y+j)$ in channel $c$; $K(i,j,c)$ is the value of the filter at position $(i,j)$ in channel $c$; and $b_i$ is the bias term associated with filter $c$.

The output $A(x,y)$ of this operation is named Feature Map, resulting from the application of the activation function to $Z(x,y)$, which is the output of the convolution layer to the $K$ filter. Which provides information about the corners, edges, or the features of the image and is read by other layers so that they can learn the remaining features of the image (*Szegedy et al., 2015*), such as illustrated in Fig. 4.

The convolution applied to image pixels automatically allows the identification and analysis of complex visual information. In contrast, support vector machines (SVMs), among other traditional techniques, typically require manual or semi-automated methods to extract features as an implementation of border filters, texture filters or histogram (*Myagila & Kilavo, 2022*, *Chaudhari, 2018*). Moreover, CNNs employ the convolution operation to mitigate their sensitivity to various image distortions and transformations, enhancing their performance in image recognition tasks.

### *Activation layer ReLU*

Activation functions $f(.)$ are commonly used after each convolution layer to insert a degree of linearity into the neuron output since the image data are not linearly separable (*Wang et al., 2020*). ReLU, represented by Eq. (3), is a widely used activation function considered among the most efficient for deep CNN.

$$\mathrm{ReLU}(x) = \begin{cases} 0, & \text{if} \quad x < 0 \\ x, & \text{if} \quad x \geq 0 \end{cases} \tag{3}$$

Despite its appearance as a linear Function, Relu (as shown in Fig. 5) actually has a derivative function which enables backpropagation. However, when the input approaches zero or falls below it, the gradient of the function becomes zero, thereby hindering the ability of the network to perform backpropagation (*Theodoridis, 2020*; *Limão, de Araújo & Frances, 2023*). Unlike other nonlinear functions with bounded output values such as zeros, positives, and negatives, the Relu has no bounded outputs other than its negative input values (*Cha, Choi & Büyüköztürk, 2017*).

In practice, the outputs of the filters are submitted to the activation function at the end of each convolutional layer. After going through this process, they are used to calculate the errors and update the neural network weights.

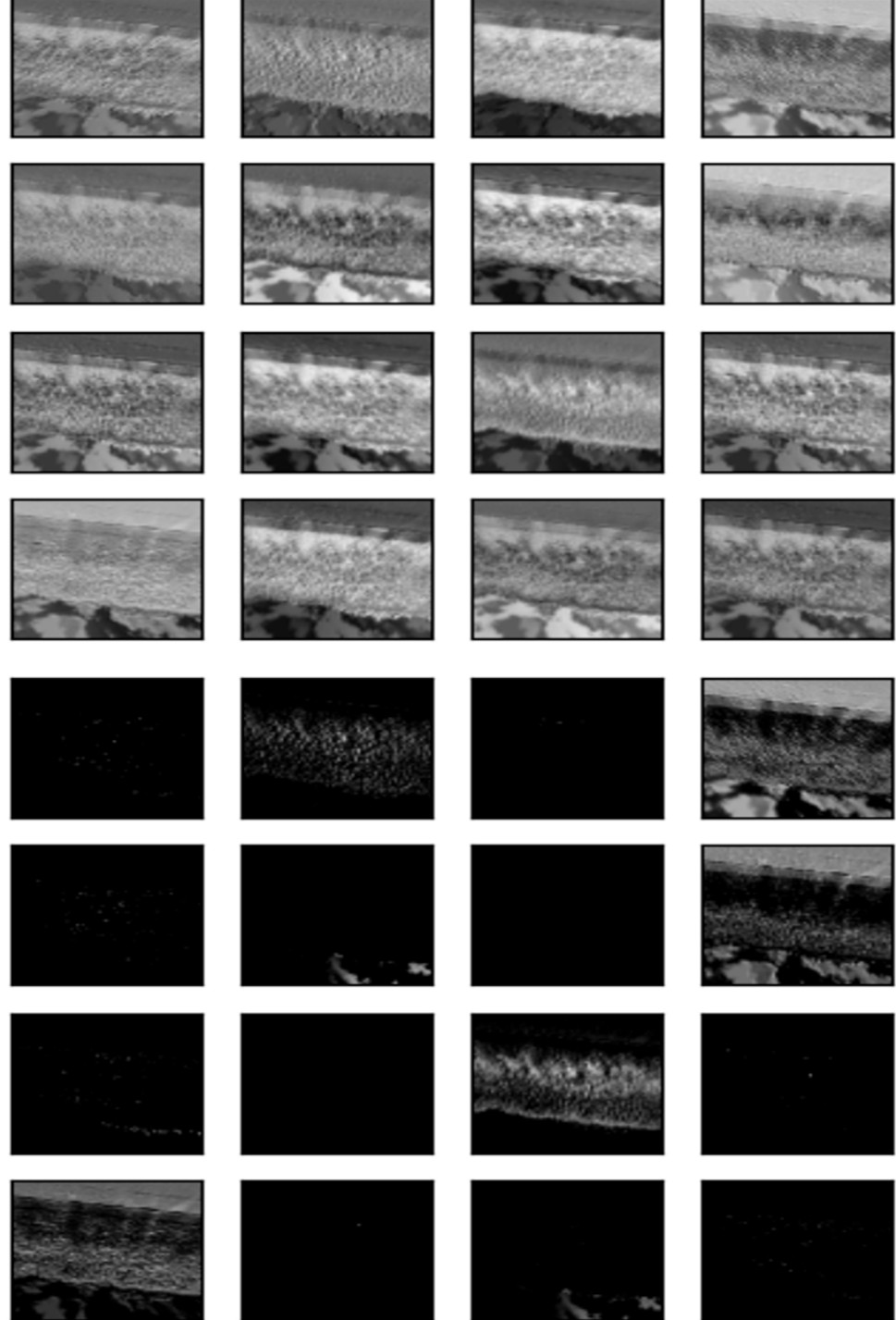

**Figure 4  Resultant feature map of each filter on image.**

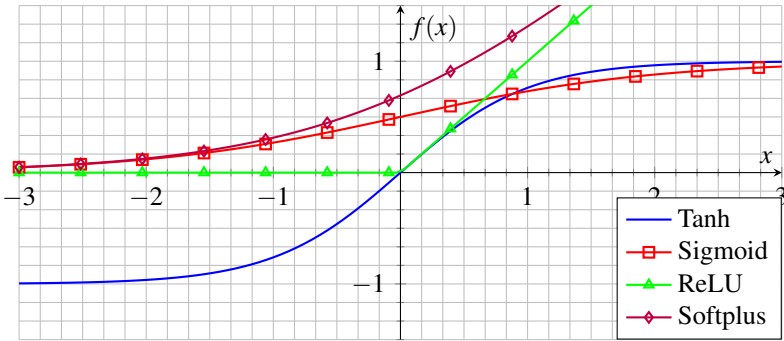

**Figure 5** Common activation functions in CNNs.

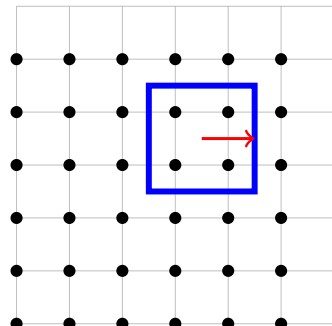

Original Image (represented by circles)
Pooling Window (Blue Rectangle)
Maximum Pixel (Red Arrow)

**Figure 6** Pooling operation on image pixels.

### Pooling layer

In general, images have a series of redundant information. Therefore, it is necessary to use the pooling layers inserted between several of the convolutional layers of the network to avoid substantial performance degradation (*Akhtar & Ragavendran, 2020*). It is a process over the convolution layer. Then, the pooling layer resizes the feature maps that resulted from the previous convolutions for the sake of dimensionality reduction. For the last step, these maps are transformed into vectors by the fully connected layer (*Li et al., 2018*).

The most suitable pooling technique for processing representations reliant on count statistics has consistently been identified as pooling. This selection is justified by its capacity to reduce the dimensionality of hidden layers by an integer multiplicative factor (*Limão, de Araújo & Frances, 2023*). This, in turn, enhances the performance in applications that entail the processing of a substantial number of images (*Murray & Perronnin, 2014*).

The Max Pooling operation involves sliding a window (often referred to as a kernel or filter) over the input image in a specified stride, and for each window position, selecting the maximum value within that window, as illustrated in Fig. 6. A pooling window (blue

rectangle) is applied to a group of pixels in an image represented by filled circles. The red arrow indicates the maximum pixel within the pooling window.

The technique achieves position invariance over larger local regions and downsamples the input image by a factor of $P$ along each axis. This is formally defined by Eq. (4).

$$\text{Max\_Pooling}(A, x, y) = \max_{p,q} A[x \cdot P + p, y \cdot P + q] \tag{4}$$

where $A(x, y)$ represents the input feature map, $x$ and $y$ indicate the spatial coordinates within the feature map. The pooling window, with a size of $P \times P$, is moved across the feature map. For each position $(x, y)$, the max pooling operation selects the maximum value within the window from the feature map $A$. This process reduces the dimensions of the input feature map by a factor of $P$ along each axis, resulting in the output pooling map. In this context, the stride $S$ is set equal to the size of the pooling window $P$ to ensure that there is no overlap between adjacent windows. Consequently, at each step, the pooling window moves $P$ units horizontally and $P$ units vertically. Leading to a faster convergence rate by selecting superior invariant features, which improves generalisation performance (*Nagi et al., 2011*).

### Dropout

Due to complicated coadaptations, training a network with a large number of neurons frequently leads to overfitting. This problem occurs when a network successfully categorises a training data set but fails to generalise, leading to poor validation and testing data performance. Overfitting has frequently been a problem in the field of machine learning (*Cha, Choi & Büyüköztürk, 2017*).

Dropout layers are added to overcome this problem. The primary objective of dropout is to randomly disrupt connections between neurons in interconnected layers with a pre-defined dropout rate (*Garbin, Zhu & Marques, 2020*). Therefore, minimising these coadaptations allows a network to generalise training samples more effectively.

During training in a CNN layer, the dropout output ($h_{\text{drop}}$) is obtained by element-wise multiplication with a dropout mask $r(x, y)$. This technique applies the dropout mask to the result of the max pooling operation, effectively deactivating a random subset of elements and reducing their activations, as represented by Eq. (5).

$$h_{\text{drop}}(x, y) = r(x, y) \cdot \big(\text{Max\_Pooling}(A, x, y)\big). \tag{5}$$

The dropout mask is a matrix of the same size as the output feature map, where each element $r(x, y)$ represents a randomly sampled value (0 or 1) drawn from a Bernoulli distribution. The probability of each element being 1 is denoted by $d$ (dropout rate), which controls the overall proportion of neurons to be deactivated during training.

$$h_{\text{drop}}(x, y) = \begin{cases} r(x, y) \cdot \big(\text{Max\_Pooling}(A, x, y)\big) & \text{com probabilidade } 1 - d \\ 0 & \text{com probabilidade } d. \end{cases} \tag{6}$$

During CNN layer inference, the output $h_{\text{inference}}$, Eq. (6), s obtained by scaling down the convolutional operation result of Max\_Pooling with a factor of $(1 - d)$ to account for dropout probability $d$ and ensure consistency with the training phase. The dropout mask

$r(x,y)$ is not applied during inference, and instead, the scaling factor $(1-d)$ adjusts the activations.

### Flatten and fully connected layers

The flattened layer converts the multi-dimensional feature maps from previous convolutional layers, including those processed by dropout, into a one-dimensional vector, as represented as Eq. (7).

$$\text{Flatten}(\mathbf{h}_{\text{drop}}) = \mathbf{X}_{\text{flat}}, \tag{7}$$

where $\mathbf{h}_{\text{drop}}$ is the input feature maps from convolutional layers of image $I$, and $\mathbf{X}_{\text{flat}}$ is the flattened output vector.

The flattened layer restores the input feature maps originating from the convolutional layers of an image into a linear vector, thereby facilitating further processing in fully connected layers.

The fully connected layer performs a crucial task in neural networks by conducting a matrix multiplication of its input with a weight matrix, which is then subjected to an activation function (*Basha et al., 2020*; *Jayanthi & V. Murali Krishna, 2022*). A single neuron in a fully connected layer can be expressed as Eq. (8).

$$Y_i = f\left(\sum_{j=1}^{N} a_{ij} \cdot X_j + b_i\right), \tag{8}$$

The $Y_i$ represents the output of neuron $i$ in the fully connected layer. $f$ is the activation function. $a_{ij}$ is the weight that links neuron $i$ to input $j$. $X_j$ represents features from the flatten layer, where $N$ is the total number of features coming from the previous layer. Finally, $b_i$ is the bias for neuron $i$.

### Adam optimiser

Adaptive optimisation enhances performance by seeking the optimal weights for a neural network, as its goal is to minimise the error function (the closer to zero, the better), leading to a reduction in the overall error of the network. The Adaptive Moment Estimation (Adam) algorithm is employed to adjust the weights of a model during the training process. So, it combines the best properties of the AdaGrad and RMSProp algorithms to provide an optimisation approach that handles noisy problems (*Zhang, 2019*).

Adam is recommended in machine learning to solve problems with heterogeneous gradients effectively. To achieve the direction and magnitude of the gradient, Adam exponential moving averages of gradients (first-moment) and exponential moving averages of the gradient squares (second-moment).

Consider the hyperparameters $\alpha$ is the Learning rate, $\beta_1$ is an exponential decay factor for the first-moment estimate, $\beta_2$ is an exponential decay factor for the second moment estimate, and $\epsilon$ a small constant to prevent division by zero. The Adam optimiser for a CNN can be described in four steps:

Step 1: Parameter initialisation

Initially, the first-moment estimate $(m_t)$, second-moment estimate $(v_t)$ are and the iteration $(t)$ set to zero.

Step 2: Update of moment Eqs. (9) and (10)

$$m_t = \beta_1 \cdot m_{t-1} + (1 - \beta_1) \cdot g_t \quad \text{and} \tag{9}$$

$$v_t = \beta_2 \cdot v_{t-1} + (1 - \beta_2) \cdot (g_t^2) \quad \text{where } g_t \text{ is the current gradient.} \tag{10}$$

Step 3: Bias correction of moments Eqs. (11) and (12)

$$m_t = \frac{m_t}{1 - \beta_1^t} \quad \text{and} \tag{11}$$

$$v_t = \frac{v_t}{1 - \beta_2^t}, \quad \text{where } t \text{ is the current iteration.} \tag{12}$$

Step 4: Weight update

$$\theta_t = \theta_{t-1} - \alpha \cdot \frac{\hat{m}_t}{\sqrt{\hat{v}_t} + \epsilon}, \quad \text{where } \theta_t \text{ represents the parameters (weights) of the CNN.} \tag{13}$$

Therefore, the Adam algorithm principle is the update of the weights (Eq. 13) based on the corrected moments and moment estimates. It combines information about the direction and magnitude of the gradients to adjust the network weights adaptively. This process repeats for each iteration during CNN training so that weights can be adjusted accordingly to minimise the loss function and enhance the performance of the model.

### Model performance assessment

When evaluating classification models, accuracy is a widely used metric to assess how well the model performs at predicting class labels for a given dataset (*Ekundayo & Viriri, 2021*; *Adnan et al., 2022*). Accuracy is presented as a per cent score between 0 and 100%, with 100% indicating that all predictions are correct and 0 meaning that none of the predictions is correct, as shown in Eq. (14).

$$\text{Accuracy} = \frac{\text{Number of Correct Predictions}}{\text{Total Predictions}}. \tag{14}$$

Loss is assessed by assessing the disparity between the predictions of the model and the actual data labels. The primary purpose of the loss function is to measure the precision of the predictions of the model (*Ekundayo & Viriri, 2021*; *Adnan et al., 2022*). Minimising the loss throughout the training process is imperative by aligning the predictions of the model as closely as possible with the actual labels. One widely employed loss formula for binary classification problems is binary cross-entropy, which can be represented as Eq. (15).

$$\text{Loss (Binary Cross-Entropy)} = -\frac{1}{N} \sum_{i=1}^{N} \left( y_i \log(p(y_i)) + (1 - y_i) \log(1 - p(y_i)) \right). \tag{15}$$

In Eq. (15) $N$ is the total number of examples in the dataset, $y_i$ is the true label, for example, $i$, and $p(y_i)$ is the probability predicted by the model that example $i$ belongs to the positive class.

The precision (P), recall (R), and F1-score metrics evaluate detection performance based on the results of predictions. Algorithm detection results can be categorised as True Positive (TP) for correct detection, False Positive (FP) for incorrect detection of nonexistent objects or misplaced detection, and False Negative (FN) for undetected ground-truth bounding boxes (*Han et al., 2022*; *Adnan et al., 2022*).

Precision, described by Eq. (16), is the ratio of true positives to all predicted positives made by a model.

$$\text{Precision} = \frac{\text{True Positives}}{\text{True Positives} + \text{False Positives}} = \frac{\text{True Positives}}{\text{all detections}}. \tag{16}$$

In Eq. (17), the recall measures the proportion of true positive predictions among all actual positive instances.

$$\text{Recall} = \frac{\text{True Positives}}{\text{True Positives} + \text{False Negatives}} = \frac{\text{True Positives}}{\text{all ground truths}}. \tag{17}$$

The F1-score, Eq. (18) is a metric that balances Precision and Recall and is used for evaluating both simultaneously.

$$\text{F1-Score} = \frac{2 \cdot \text{Precision} \cdot \text{Recall}}{\text{Precision} + \text{Recall}}. \tag{18}$$

An examination of recall information, precision, and F1-Score can yield valuable insights into the functionality of the model, particularly with regard to false positives, false negatives, and true positives. The confusion matrix is a powerful tool for visually representing how the model categorises samples into different categories (*Azimi & Pekcan, 2020*).

Furthermore, the ROC curve visually represents how a model predicts true labels in the binary classifier. The AUC metric measures the overall quality of the ROC curve and briefly summarises the ability of the model to distinguish between different classes at different classification thresholds (*Azimi & Pekcan, 2020*). The larger the area under the curve, the better the model performs. An AUC score of 1 means perfect classification, while a score less than 0.5 indicates random selection (*Adnan et al., 2022*; *Cano et al., 2021*; *Azimi & Pekcan, 2020*).

## OVERVIEW OF THE PROPOSED METHOD

Visual inspection is the primary method for detecting failures in geotechnical structures, but on-site visits can be risky due to unstable soil. Professionals can rely on images to identify defects. Inspections using drones before identifying damages and maintaining compromised areas can be helpful because they reduce human exposure to unsafe conditions. Binary image classifiers can also monitor surface faults, as demonstrated in a similar method (*Limão, de Araújo & Frances, 2023*). For this study, instead of using an internet dataset, we collected local images to detect damaged areas. In addition, we use data augmentation techniques, both geometric and photometric, to improve the data.

This section summarises the entire process of our framework of the purpose. Figure 7 depicts the overall methodology, which encompasses the stages of data acquisition, pre-processing, training, and testing. To train a CNN classifier, raw images of damaged surfaces

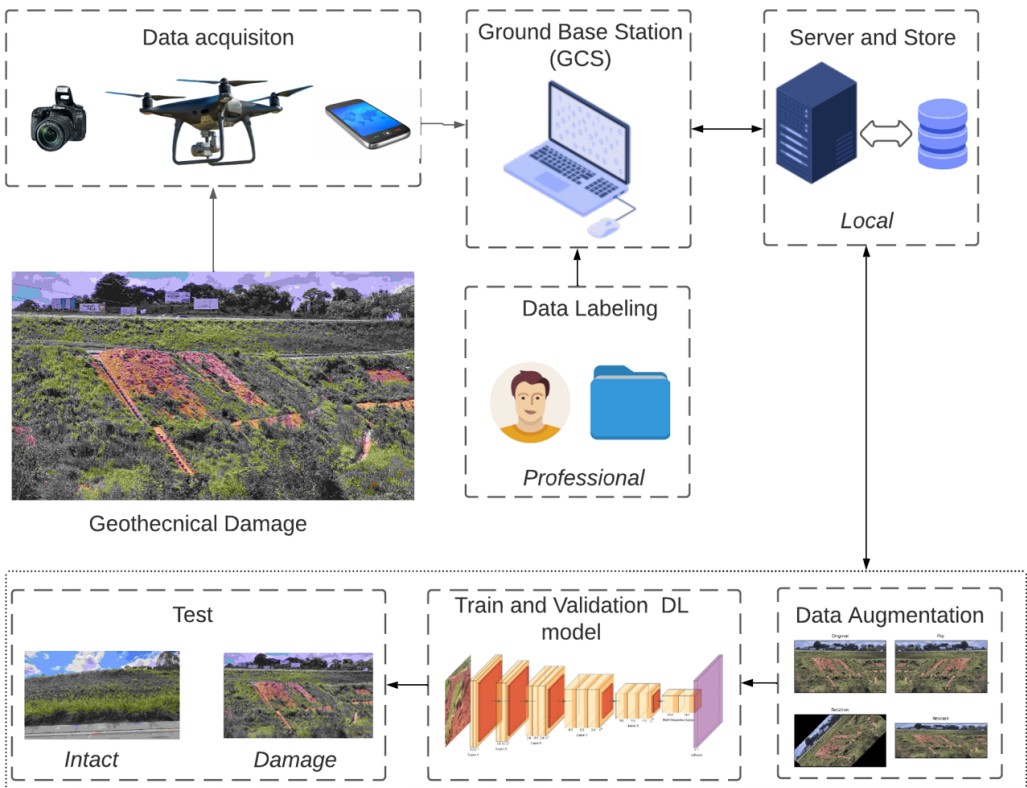

**Figure 7** Flowchart for detecting surface failure on geotechnical structures.

were captured using cameras and devices, including cellphones and UAVs. The image dataset includes examples with a broad range of variations, such as lighting and shadows, that can potentially trigger false alarms.

In this article, damage is defined as visually perceptible in images by the naked human eye, without the aid of magnification or additional tools. Negative dataset examples include images of landslides, without ground cover and erosion, whereas negative ones do not. Figures 8 and 9 illustrate the definition.

# BUILDING CLASSIFIER FOR DETECTING SURFACE DAMAGE ON GEOTECHNICAL STRUCTURES

This section explains the factors considered when creating the dataset and setting the hyperparameters used to train the CNN model. Being an empirical process, there are no precise rules for hyperparameter optimisation, making it difficult to configure and select suitable values (including learning rates and regularisation parameters). As a result, the selection of the best network design for this damage detection is determined by trial and error while directed by the error of the validation set (*Bengio, 2012*).

For the environment setup of this work, the following technologies were used:

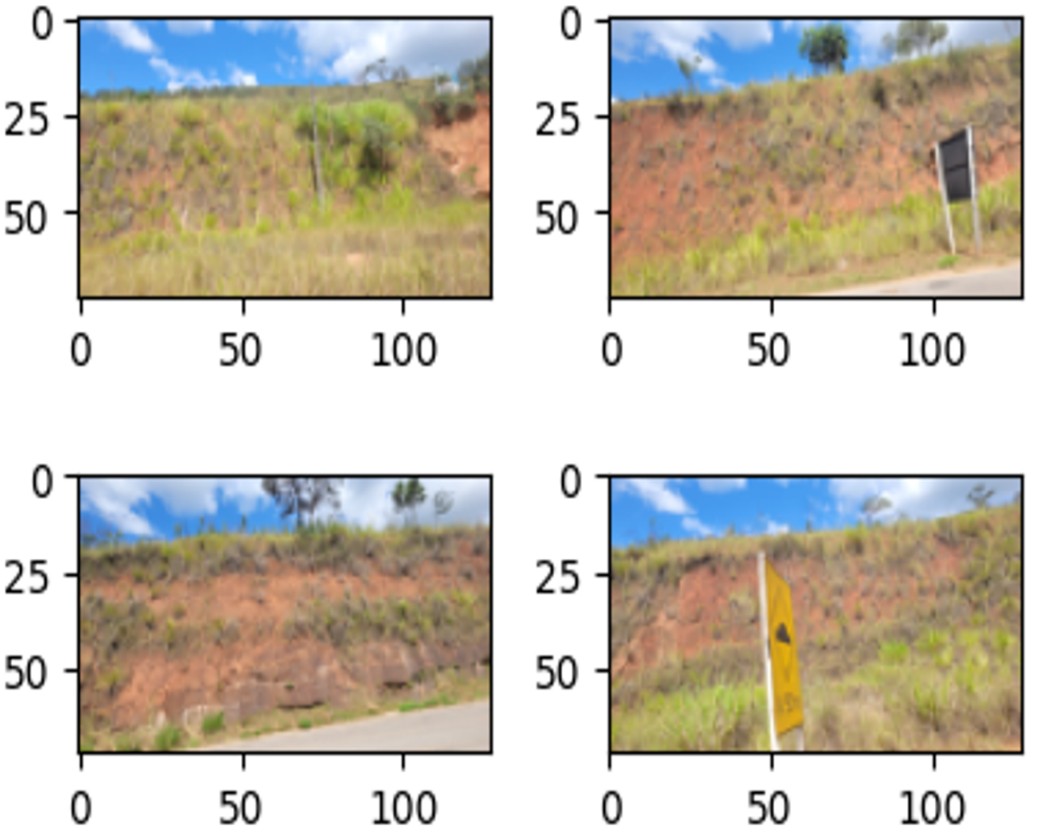

**Figure 8** Examples of images in the training set labeled as 'Damage'.

- Hardware: Windows 11 Home x64, 16GB RAM and Intel(R) Core(TM) i7-10510U CPU @ 1.80 GHz 2.30 GHz.
- Programming language: Python 3.9.16.
- Web-based interactive computing platform: Project Jupyter.
- Main Python Libraries: TensorFlow 2.8.0, Keras 2.12.0, Pandas 1.4.1, NumPy 1.22.3, Matplotlib 3.5.1 and Scikit-learn 1.0.2.

## Dataset generation

Public databases contain images of various types of geotechnical damage (*American Geosciences Institute, 2024*; *Nevada Bureau of Mines and Geology, Acess em 29 de January 2024*; *GeoNet, 2024*). However, these databases have limitations in terms of the quantity and diversity of available images. We found no representative images (in terms of quantity and quality) in the aforementioned databases that could potentially represent surface damage on slopes (*Kattenborn et al., 2021*). For instance, landslides and erosion are found in different categories of geotechnical structures like volcanoes and groundwater, which are not the focus of our study. Therefore, a significant effort is required in searching and labelling the images so that the public databases are consistent with the problem we aim to address with the help of computer vision.

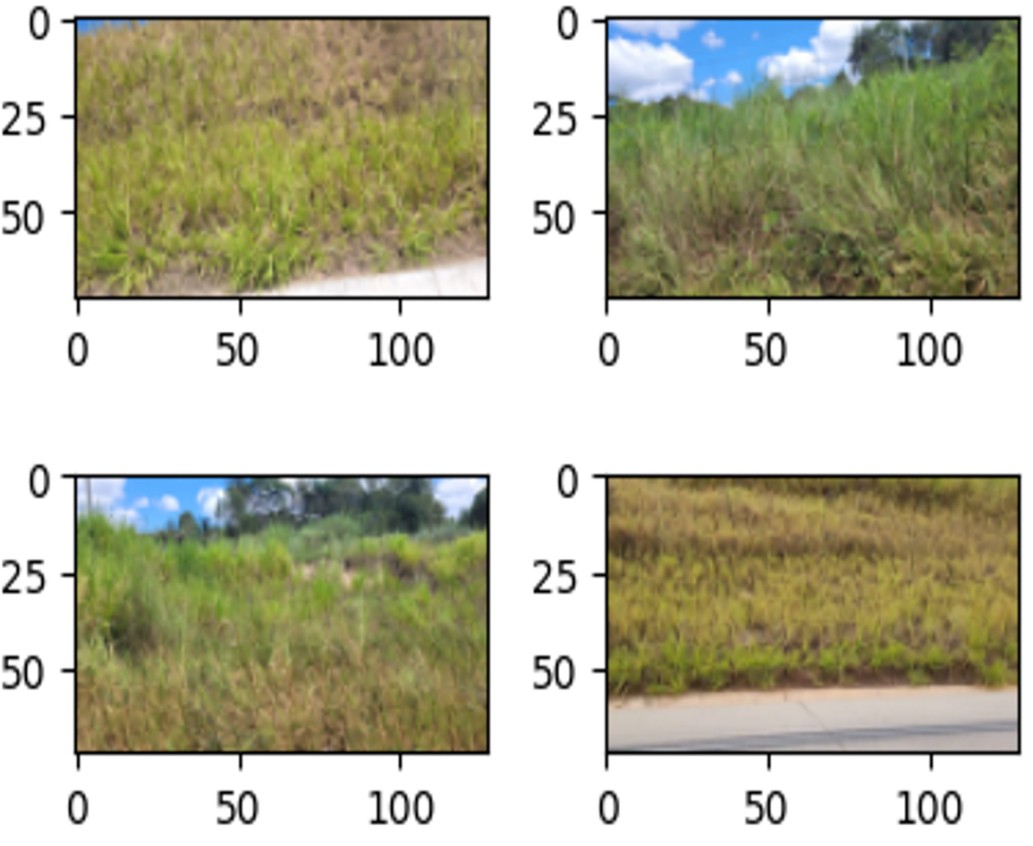

**Figure 9** Examples of images in the training set labeled as 'Intact'.

Additionally, most of the images in these databases are from satellites, which has some challenges (*Zhong et al., 2020*; *Solórzano et al., 2022*). For instance, accessing these images can be quite expensive, and users may need to subscribe or purchase them. Also, weather conditions such as clouds, fog, and precipitation can significantly affect the image quality and limit their utility at certain times of the year. Furthermore, the limited resolution of the images may not represent the specific ground conditions, and the users have no control over capturing the images.

Therefore, we decided to use our dataset for a real and authentic case study. Our approach involves using emerging technologies like UAVs embedded with cameras or mobile devices. This approach allows users to have greater control over the data acquisition process and access more accurate images (*Kanellakis & Nikolakopoulos, 2017*; *Greenwood et al., 2019*).

Images were collected from landslides, without ground cover and superficial erosion on 381 national Brazilian roads (close to João Monlevade city) from January to May 2022 (*De Araujo, 2023*). A period with the highest number of recorded landslides in the last 23 years in Minas Gerais, Brazil, *Macedo & Sandre (2022)*. A total of 337 RGB (traditional red, green, blue) images were captured using mobile cellphone cameras and a camera attached

to a UAV, and manually annotated as 'damage' or 'intact' images, as illustrated in Figs. 8 and 9.

Furthermore, although geotechnical monitoring data has grown with development, compared to other fields, it remains insufficient for full deep learning applications (*Phoon & Zhang, 2023*; *Zhang, Li & Li, 2021*). The number of examples (422 images) in the dataset is limited, which can compromise CNN performance and result in low model accuracy, *i.e.,* overfitting. If the model only learns from a few examples of a given class, it is less likely to make correct predictions on unseen data (*Shorten & Khoshgoftaar, 2019*). Then, the training dataset underwent data augmentation (DA) techniques, which introduced randomised geometric and pictorial transformations to each image. This approach ameliorates the challenge of limited data by providing additional invariance to the training dataset (*Tian et al., 2023*), and the dataset now contain 674 images, which is an increase in size. Vision-based approach for detecting cracks on concrete images was proposed using a deep learning method.

Afterwards, as part of the pre-processing phase for the image analysis, it was necessary to normalise the pixel values within a range of 0 to 255 for a range of 0 to 1. By standardising the pixel values, we can ensure that the resulting data is consistent and reliable for further processing.

### *Augmentation*

The image transformations can be based on movement concerning the axis of the image (geometric), changes in the shape of elements, size, and position within the image, or act on their pictorial composition (photometric). For example, flipping, rotating, cropping, shearing, rescaling and zooming are geometric DA, and noise injection, colour jittering, random erasing, and principal components analysis are photometric DA.

In this problem, the number of images in the data set of the train is duplicated (337 to 674 images). The geometric properties of flipping, rotating, and rescaling are applied only to the training dataset. In addition, we applied the pictorial composition transformation-based augmentation by a smoothing filter in the image. This process improves model generalisation ability when dealing with new data.

The objective of the data augmentation was to expand and diversify the images in the training set to improve the performance of the model (*Ekundayo & Viriri, 2021*). Figure 10 shows some examples of transformed images.

### *Sample division*

A total of 674 images are randomly selected from the dataset to generate training, validation and testing sets. The pre-processed set with augmentation and training set images is fed into a CNN to build a binary classifier that can distinguish between 'damage' (negative output, probability close to 0) or intact (positive output, close to 1). The composition of the dataset includes training, validation, and test data, as shown in Table 2.

## Model architecture

Low-complexity CNN models demonstrate comparable or better performance than state-of-the-art architectures in scenarios with limited training samples (*Brigato & Iocchi, 2021*;

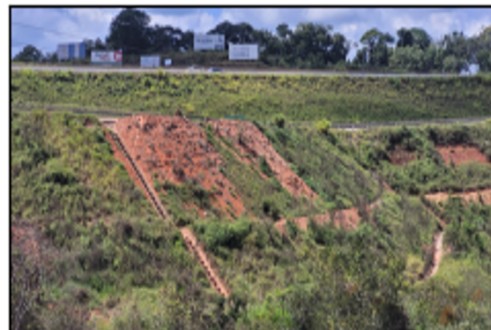
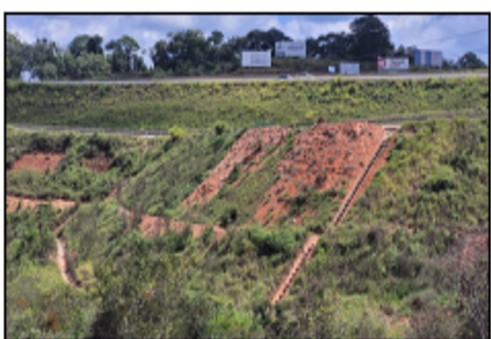
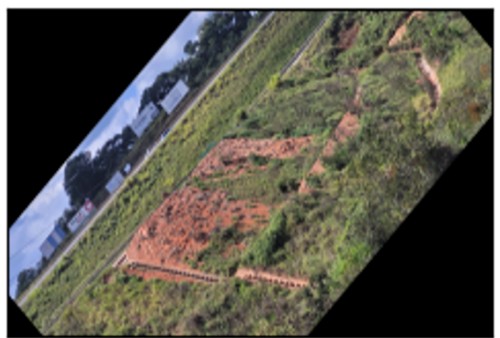
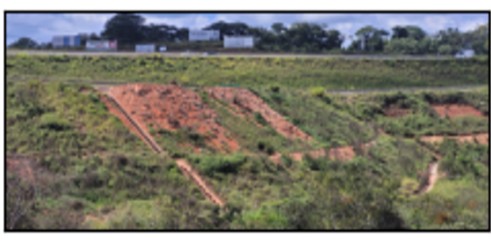

**Figure 10** Effects of geometric data augmentation on the image.

**Table 2  Sample division data.**

| Class | Sample division | | | | | |
|---|---|---|---|---|---|---|
| | Training | | Validation | | Testing | |
| | Images | (%) | Images | (%) | Images | (%) |
| Damage | 312 | 46.29 | 53 | 7.86 | 53 | 7.86 |
| Intact | 192 | 28.49 | 32 | 4.75 | 32 | 4.75 |
| Total | 504 | 74.78 | 85 | 12.61 | 85 | 12.61 |

*Foroughi, Chen & Wang, 2021*; *Limão, de Araújo & Frances, 2023*), as with our problem. For this reason, we perform a custom CNN architecture that can be created using multiple layers: input, convolution, pooling, activation, and output. Convolution blocks comprise a sequence of convolution, pooling, and dropout layers. The code is available at *Araujo (2023)*.

Regarding the number of connected layers, as the complexity and size of networks increase (depth, width and resolution), considerable computing resources are necessary for training and evaluation since the number of parameters also increases considerably (*Li et al., 2022*; *Darapaneni, Krishnamurthy & Paduri, 2020*). Due to the hardware constraints

**Table 3  Comparison of average metrics among various architectures.** The bold represents the two best values of the metrics (highest accuracy, lowest loss) for the evaluated architectures.

| CNN model | Training | | Validation | | Testing | |
|---|---|---|---|---|---|---|
| | Accuracy(%) | Loss ($\times 10^{-3}$) | Accuracy(%) | Loss ($\times 10^{-3}$) | Accuracy(%) | Loss ($\times 10^{-3}$) |
| 32 filters. 1 FC 128 neurons | 94.26 | **15.17** | 90.00 | 22.88 | 87.06 | 53.69 |
| 32 filters. 1 FC 64 neurons | **94.65** | 15.87 | 90.00 | 33.65 | 84.71 | 44.56 |
| 32 filters. 2 FC 64 neurons | 88.71 | 26.96 | 90.00 | 26.41 | 80.00 | 58.95 |
| 32 filters. 2 FC 128 neurons | 94.26 | 15.62 | **94.00** | 24.72 | **92.94** | **26.38** |
| 64 filters. 1 FC 64 neurons | 61.98 | 66.47 | 66.47 | 66.31 | 62.35 | 66.31 |
| 64 filters. 1 FC 128 neurons | 94.06 | 15.57 | **94.00** | 20.36 | **90.59** | **25.86** |
| 64 filters. 2 FC 64 neurons | **95.45** | **11.66** | 90.00 | 25.11 | 89.41 | 52.80 |
| 64 filters. 2FC 128 neurons | 86.53 | 33.56 | 84.00 | 40.77 | 76.47 | 47.99 |

of the experiments (single CPU), it is justified to conduct trials with 1 to a maximum of 3 layers in order to improve model performance.

The input layer is the top layer with a $128 \times 128 \times 3$ resolution. Each dimension denotes the height, width, and channel (red, green, and blue). The max pooling performance in image data sets is better than that of mean pooling (*Scherer, Müller & Behnke, 2010*), so max pooling is justified. The dropouts are fixed and can be implemented within the aforementioned layers according to their intended purposes.

As mentioned above, this paper used the Adam optimiser, which is widely considered the best optimiser for deep-learning scenarios. The capability of the Adam optimiser to adapt its learning rates addresses the challenge of gradient disappearance (*Li et al., 2022*), which is often observed in low-complexity CNN architectures.

Using the sigmoid activation function, the output layer predicts whether each input data is a damaged or intact surface after the convolution processes (binary classification). A sigmoid function maps real numbers to values between 0 and 1, making it useful in binary classification (*Wang et al., 2020*). After going through the sigmoid activation on the last layer, the CNN outputs a number between 0 (0%) and 1(100%), indicating the probability of each class.

Our selection of architecture was based on the average accuracy and loss model from five model runs, ensuring a fair performance comparison between the architectures and minimising the impact of random factors such as weight initialisation and training variability. Our previous tests on two-layer convolution models showed average worst accuracy and losses, prompting us to evaluate the metrics for the three-layer convolution model. Table 3 reports the performance metrics for several different architectures In bold are the two best values of the metrics (highest accuracy, lowest loss) for the evaluated architectures, provided to aid visualization and selection of the architecture with the best results.

The CNN architecture proposed for the detection of geotechnical damage is illustrated in Fig. 11.

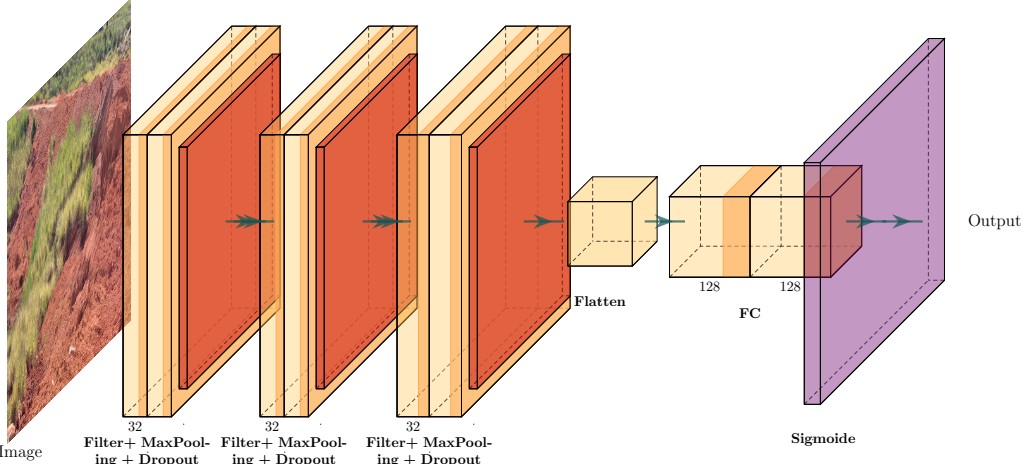

**Figure 11** **The custom CNN structure that achieved the best average performance, adapted from _Iqbal (2018)_.**

# EXPERIMENT RESULTS AND DISCUSSION

To validate the model, we employed a specific architecture consisting of three convolutional layers, with a dropout rate of 0.25 in layers 2 and 3, and two fully connected layers containing 128 neurons each and utilised a sigmoid activation function. This combination was selected due to its superior average performance in terms of accuracy, loss during training, validation, testing, and training time compared to models employing 64 filters in the hidden layer.

Comprehensive testing was conducted to evaluate the performance of the proposed network architecture. It is important to note that our team conducted a rigorous testing process, which included a minimum of five training runs for each architecture and corresponding hyperparameters, as shown in Table 3. After the training and validation run to compute the accuracy and loss metrics, as shown in Figs. 12 and 13, respectively.

The outcome of the filter process is performed in an image by each convolution layer in the form of a feature map. To illustrate the feature map of 16 of the 32 filters applied to each convolutional layer, the image shown in Fig. 14 was fed into the input layer of the network. Architectures of 16 and 32 filters, each with a size of 3x3, are commonly used in CNN for computer vision (_Bari et al., 2021_; _Cha, Choi & Büyüköztürk, 2017_; _Li et al., 2019a_).

The subsequent images Figs. 15, 16, and 17 represent the activation maps that are generated after the convolution operations. The features generated by these convolutions become increasingly complex as the information is propagated through the network. CNNs extract features from raw image pixels, while traditional algorithms (SVM, random forest, among others) require pre-defined features as input (_Affonso et al., 2017_; _Lei et al., 2020_). In the images of Figs. 15, 16, and 17, the yellow values are negative and represent the characteristics detected by each kernel(features). On the other hand, higher values indicate

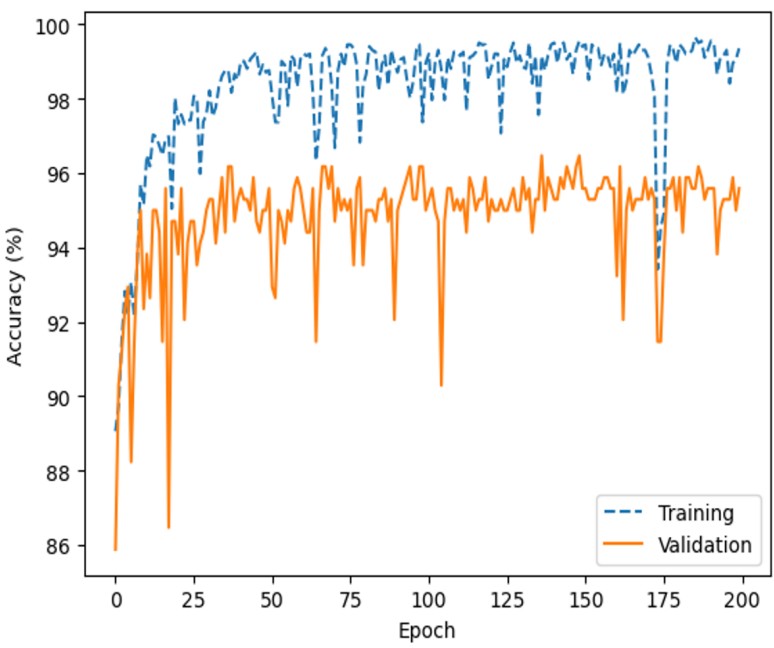

**Figure 12 Accuracy of the average performance.**

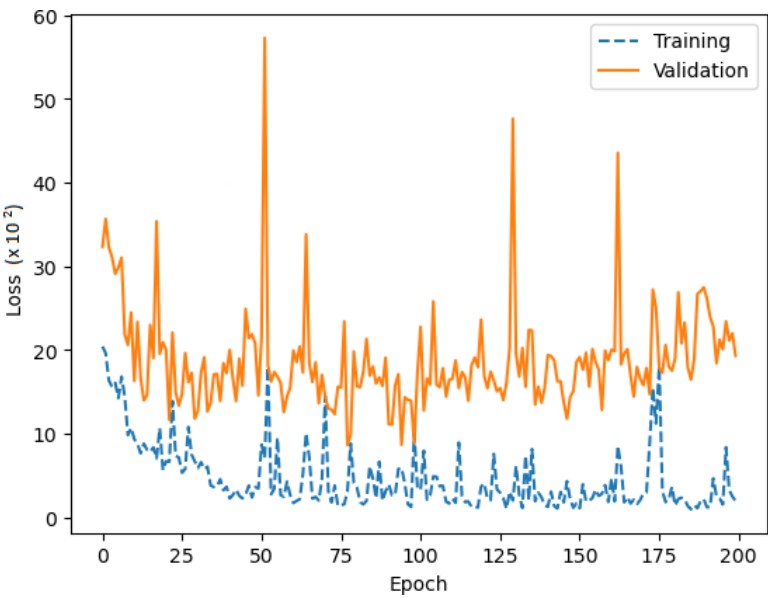

**Figure 13 A loss of the average performance.**

that the area of the image corresponding to them has a high level of significance to the description of the positive ('intact') class.

Although there are several superficial damages on slopes that several classes may represent, the database limited the model to a binary classifier. Thus, the sigmoid function

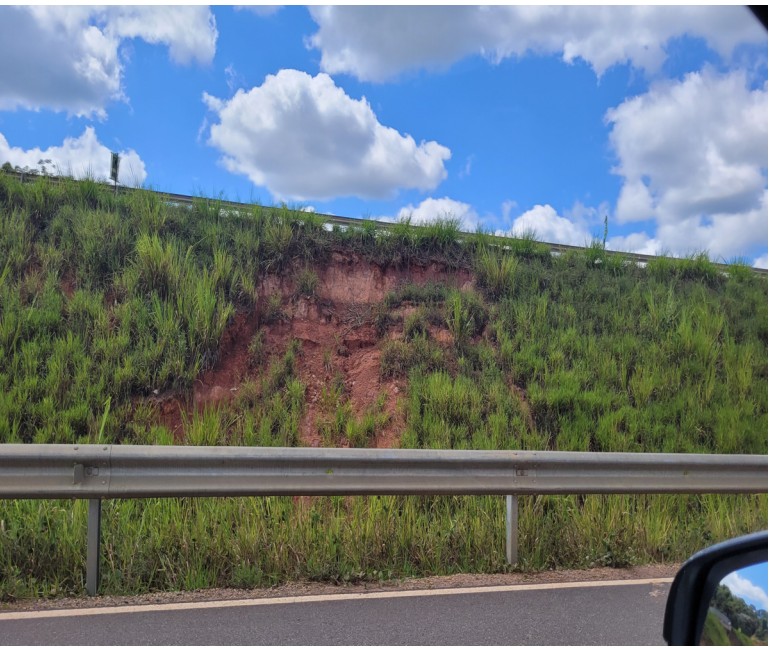

**Figure 14** Input image.

generates output values ranging from 0 to 1, representing the probabilities of each class. We used a 0.5 threshold to delimit the two classes. Therefore, if the output value is less than 50.00% (probability less than 0.5), the image is classified as 'damage'. Images with probabilities equal to or greater than 50.00% are classified as 'intact'. The detection rigidity can be increased by adjusting the threshold. Higher threshold values indicate a smaller range of intact regions. Examples of the test set were fed into the network, and their respective outputs are presented in Fig. 18.

In order to describe and compare the performance of the classification models during the testing phases, the confusion matrix and ROC curve are provided (*Cano et al., 2021*). Figures 19 and 20 illustrated the performance results.

The highest average accuracies in training and validation are 94.26% and 92.00% less than the 50th epoch, respectively. Additionally, the capability of the model to distinguish between the two classes, 'intact' or 'damage', is excellent, as indicated by the ROC curve and an AUC score of 0.99 in Fig. 20. The proposed architecture demonstrated satisfactory performance, and state-of-the-art results (*Brigato & Iocchi, 2021*; *Foroughi, Chen & Wang, 2021*; *Limão, de Araújo & Frances, 2023*) were achieved on image datasets from geotechnical damage. However, to identify damage at the geotechnical surface level, one needs to explore computer vision techniques for object detection tasks because there are singular challenges in the geotechnical engineering field (*Phoon & Zhang, 2023*).

There are several categories of damages that can occur on geotechnical surfaces (*Terzaghi, Peck & Mesri, 1996*; *Salajegheh, Mahdavi-Meymand & Zounemat-Kermani, 2018*; *Solórzano et al., 2022*; *Khan et al., 2021*). UAVs and mobile devices to capture aerial images have been widely used in the field (*Kanellakis & Nikolakopoulos, 2017*; *Kattenborn et al., 2021*;

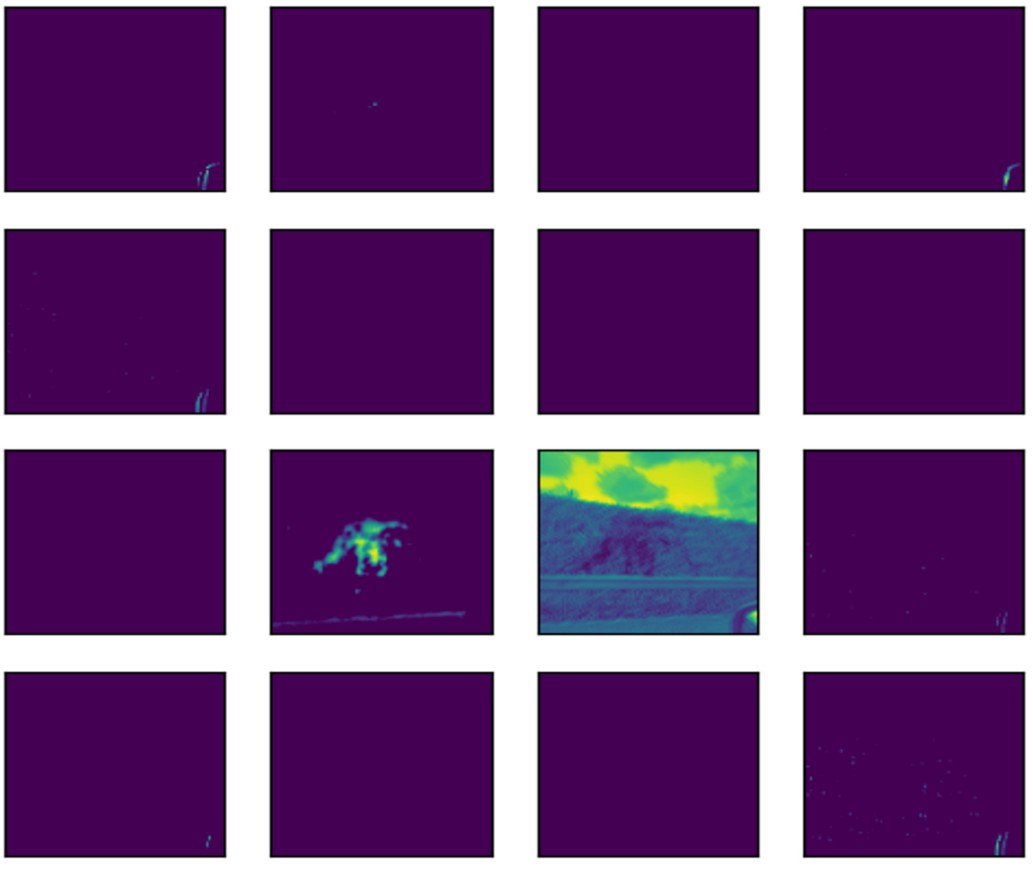

**Figure 15 Feature map of first layer.**

*Jang, Kim & An, 2019*). While the volume of data collected by UAVs and mobile devices is substantial, it can be redundant in some situations. The uniform vegetation cover can reduce dataset diversity and increase the risk of overfitting, so dataset preparation is necessary.

Labelling the images accurately required identifying and removing duplication and categorising the surface-level damages. This stage of dataset preparation resulted in a quantitative analysis of 337 images, which is considered a small size when compared to most computer vision approaches that use quantities in the order of thousands, such as studies by *Abedalla et al. (2021)*, *Wang et al. (2020)*, *Yadav & Jadhav (2019)*, *Lu, Tan & Jiang (2021)*, *Bari et al. (2021)* and *Affonso et al. (2017)*. This stage of the study combined the efforts of soil engineering experts.

Although there are public databases available (*Nevada Bureau of Mines and Geology, Acess em 29 de January 2024*; *American Geosciences Institute, 2024*; *GeoNet, 2024*), their contribution does not help construct a dataset that can be used to evaluate land cover, landslides, and erosion because they are limited in terms of both variety and size. Therefore, obtaining a representative dataset required a significant amount of effort, as previously

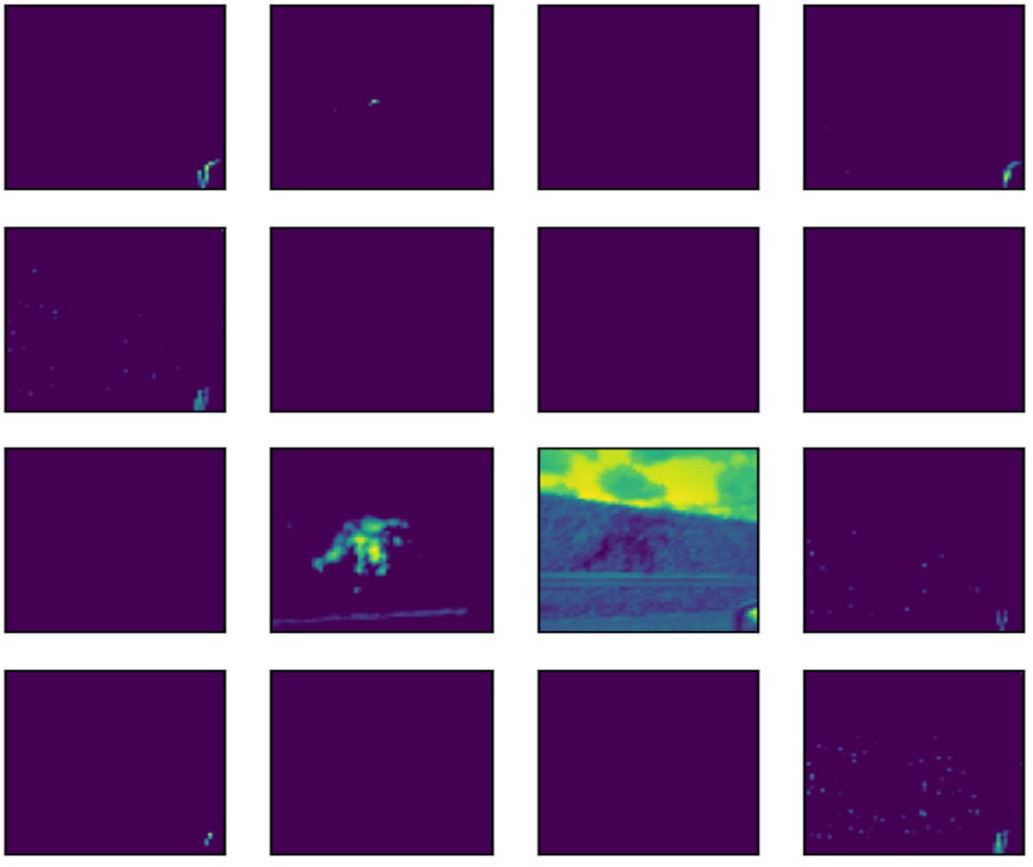

**Figure 16  Feature map of second layer.**

stated by *Kattenborn et al. (2021)*. We would like to emphasise that this dataset was curated by skilled professionals in the field.

To compensate for the limited dataset for building the database, data augmentation was used to enhance data diversity in the set, and we achieved 674 images. In image classification problems, domain adaptation has proven to be an efficient strategy for diversifying databases (*Cano et al., 2021*; *Abedalla et al., 2021*) and significantly improving classification results in small datasets (*Ottoni, de Amorim & Novo, 2022*). Additionally, we proposed a binary classifier model as a solution to a limited database or lack of images for classes. The first class contains images with visible indicators of common surface damage such as erosion, landslides, and soil without cover. The second class represents healthy soil.

Besides, there is a hypothesis that suggests that certain traditional algorithms may have the potential to produce models that are more accurate than those generated by CNN in image classification tasks (*Affonso et al., 2017*). However, the circumstances may create obstacles that could make it challenging to extract the manual feature. This makes CNN advantageous in computer vision tasks since filters in convolutional layers automatically extract features (*Di et al., 2014*; *Myagila & Kilavo, 2022*).

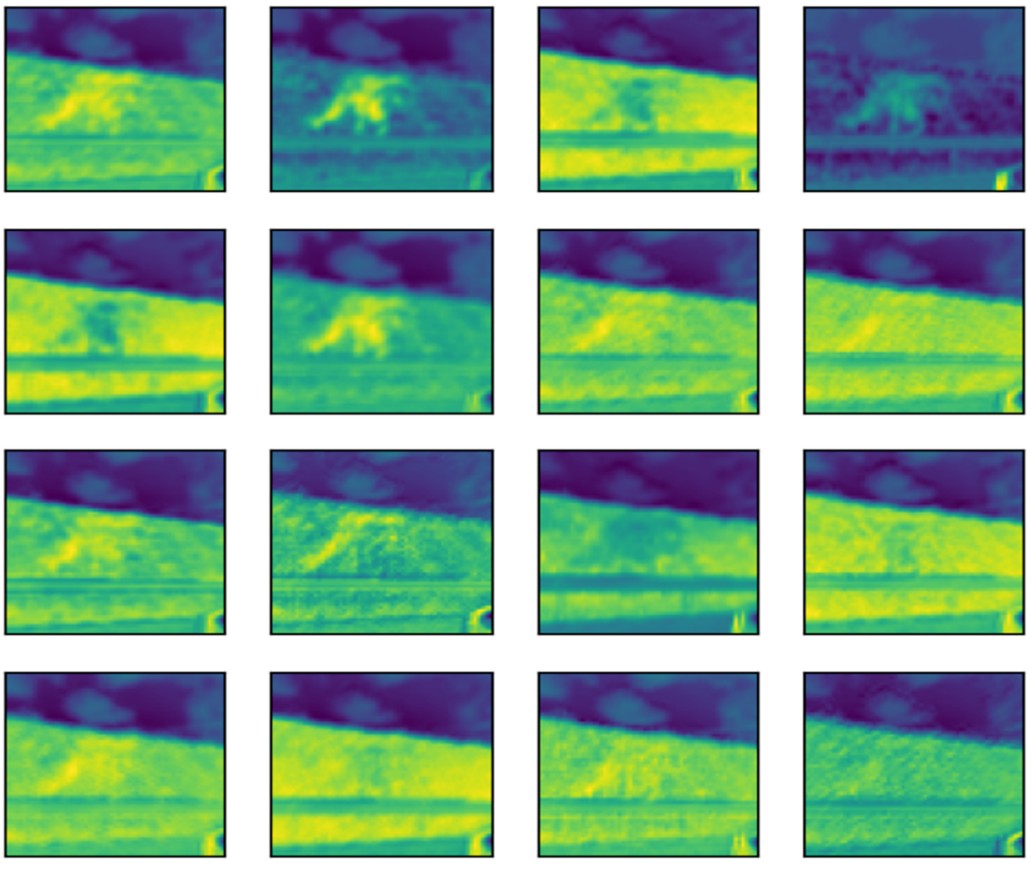

**Figure 17** Feature map of third layer.

Therefore, to validate and test the CNN model, we conducted experiments with various model configurations, including different numbers of filters in convolutional layers and neurons in the fully connected layer, Table 3. After testing several combinations, the best average accuracy and loss results were achieved with two connected layers of 64 neurons with a convolutional layer of 32 filters and one connected layer of 128 neurons with a convolutional layer of 64 filters. Regularisation methods were applied to prevent overfitting, including the dropout and the Adam optimisation, along with binary cross-entropy loss. Moreover, we decreased the training batch size of the set. Using smaller batch sizes can aid in model convergence by updating network weights more often.

In smaller datasets, changes to parameters can significantly affect gradient distribution, which can be seen in oscillation patterns on learning curves, Figs. 12 and 13. To tackle this issue, we increased exposure to examples by utilising a batch size of 6 and we used the Adam optimisation algorithm to stabilise the learning curve convergence (*Li et al., 2022*). The algorithm helps reduce the differences in weight updating that occur due to the variation of the gradient, thus softening oscillations in the learning process. In addition, the Adam optimiser resulted in the prevention of overfitting (*Shorten & Khoshgoftaar, 2019*)

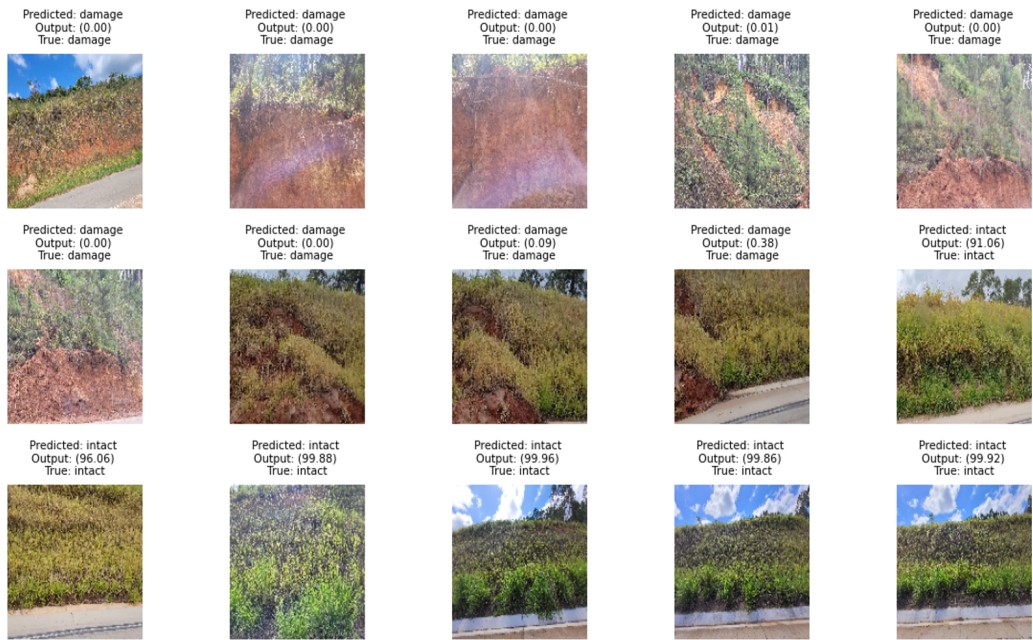

**Figure 18** Testing CNN models to detect surface failure in geotechnical structures.

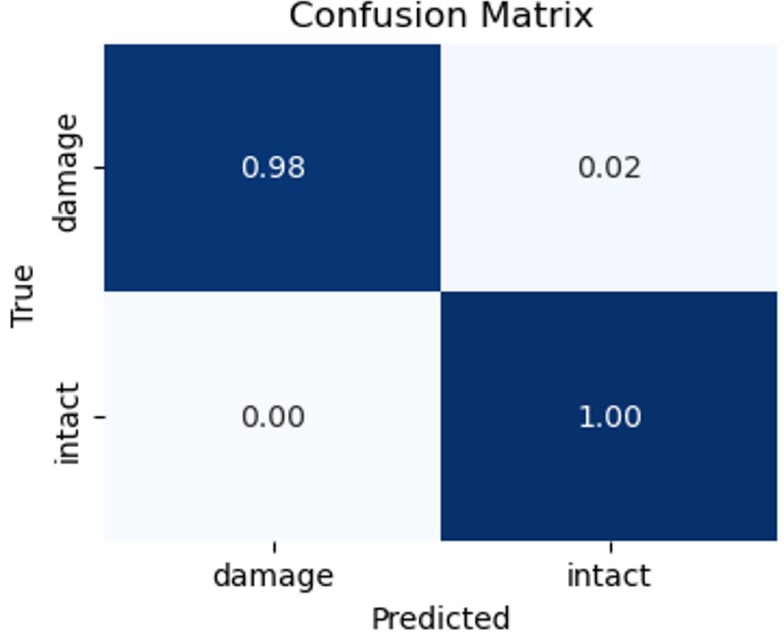

**Figure 19** Matrix confusion of the testing set.

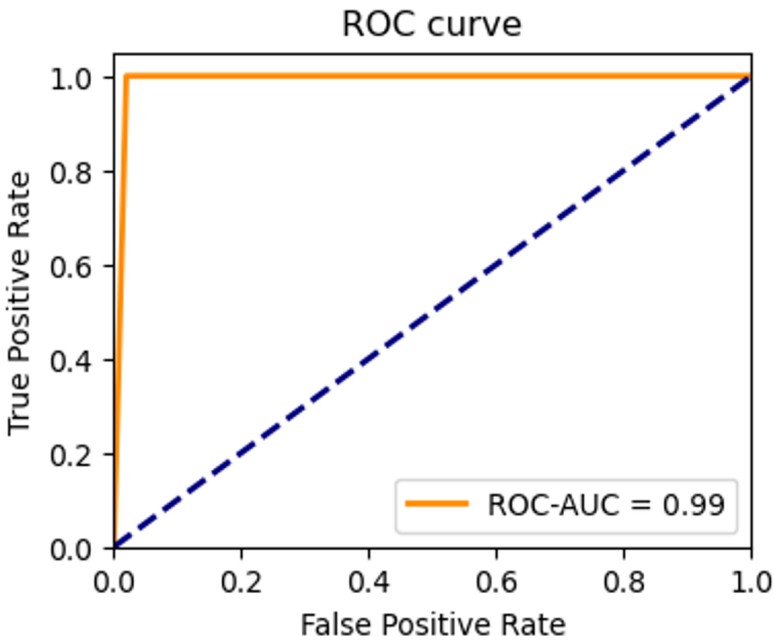

**Figure 20  ROC curve of the testing set.**

and gradient reduction (*Li et al., 2022*), a common problem with leaner CNN architectures (*Brigato & Iocchi, 2021*).

To assess the efficacy of the model, we employed the techniques of the confusion matrix and receiver operating characteristic curve with area under the curve (ROC-AUC) in the test dataset. These metrics are accepted as standard methods for evaluating the performance of predictive models of the image classification tasks (*Han et al., 2022*; *Cano et al., 2021*). The analysis of these metrics allowed us to gauge the ability of the model to classify the classes. The confusion matrix shows satisfactory results. The model correctly identifies all intact images and predicts 98% of damaged images of the testing dataset. That is, the confusion matrix results indicate that out of 100 images belonging to the 'intact' class, the model accurately classified all 100 images into the correct class. Similarly, out of 100 images belonging to the 'damage' class, 98 were correctly classified as belonging to the 'damage' class, while two were mistakenly classified as belonging to the 'intact' class. Additionally, an AUC score of 0.99 indicates that the model is excellent at distinguishing between the two classes on slope surfaces. This means that the model can accurately differentiate the positive examples (intact class) from the negative examples (such as damage caused by lack of vegetation cover, landslides, and erosion). This demonstrates the potential of the CNN model to enhance failure detection performance within the field of geotechnical sciences.

In addition to the challenges related to the build the dataset, we encountered other obstacles during the architecture experimentation. Particularly, training these CNN models incurred significant computational time. The approximate running time on only the CPU is about 14–20 h for each five running training. One of the ways to reduce training time

is through GPU-based parallel processing (*Alzubaidi et al., 2021*; *Garbin, Zhu & Marques, 2020*; *Cano et al., 2021*), which we suggest as a future implementation.

Finally, although the cost of implementing the CNN model is relatively high, the overall performance can be better than other techniques (*Girshick, 2015*). In CNNs, feature extraction techniques are unnecessary, as CNN automatically learns features from input data. This advantage can save a lot of effort in implementing pattern detection in images, which is valid for the present study. And an advantage when compared to other techniques for image processing, such as SVM. Additionally, even though the custom CNN performs well in detecting damage on geotechnical surfaces (*Cha, Choi & Büyüköztürk, 2017*; *Limão, de Araújo & Frances, 2023*), it is still important to compare its performance to popular CNN models like VGG, Inception, and ResNet. Recent studies suggest that simpler custom CNN architectures can provide comparable results to these popular models (*Brigato & Iocchi, 2021*; *Foroughi, Chen & Wang, 2021*). However, our specific problem requires further investigation and analysis.

## CONCLUSION

A computer vision-based approach for detecting surface damage in geotechnical structure images was proposed using deep learning methods. The authentic images used in creating the dataset were captured, asses UAV and mobile devices, and subsequently divided into training, validation, and test sets for the development of a CNN model. The images depict landslides, without ground cover and erosion on slopes near a Brazilian highway.

It is worth noting that the volume of images collected by UAVs and mobile devices for inspection is significant but considerably redundant for slopes. The uniform and typical vegetation coverage under healthy conditions reduces the diversity of the dataset, which can lead to the model being overfitted. When it comes to soil with visible damage to the surface, constructing a dataset with several categories (classes) of damage requires significant efforts as public domain images are limited. This makes a multiclass classification model unfeasible due to the disproportion of class size. To tackle this issue, we opted for solutions such as eliminating the images that did not represent the problem domain with the aid of a specialist. We also prepared the images (labelled and applied data augmentation techniques) and built a binary classifier. However, due to the redundancy of the images and the insufficiency of the public dataset, the database lacks more images representing failure indicators. Which was a challenger.

Therefore, we applied data augmentation techniques and regularisation in the database to reduce the common problems in model training with scarce datasets. Subsequently, we perform low complexity architectures and aim to construct a binary classifier based on convolutional neural networks to classify images from the surfaces of geotechnical structures as either 'damage' (without ground cover, landslide, erosion) or 'intact'. We tested eight different model combinations involving three convolutional layers and fixed hyperparameters, with varying numbers of neurons both in the convolutions and the fully connected layers. The best average results of accuracy and losses were the 32 filters in each convolutional layer, two FCs of 128 neurons each.

Metrics ROC curve and AUC score in the test set within the proposed CNN architecture show promising and satisfactory results when analysing failure scenarios in geotechnical areas. In the state of the art, the CNN models are considered robust in processing noisy images with inadequate or distorted illumination. Nevertheless, we can not say that the model performs well in detecting damage to geotechnical surfaces with images collected in adverse weather or lighting conditions. Therefore, to gain a better understanding of the issue, it is crucial to conduct more extensive and diverse studies. These studies should cover datasets in various situations, including images collected during extreme weather events that result in environmental disasters and tragedies.

Future research endeavours aim to train the same CNN architecture using an expanded dataset comprising thousands of images. We understand that we still need to advance and conduct studies to perform more models, including pre-trained models, parallel processing, and generating artificial data to overcome the limitations of the damage identification problems. This will assess the continued adequacy of performance and determine if refinements are necessary to enhance the accuracy of the model. The implementation of a CNN requires a substantial volume of training data to build a robust classifier. One common limitation in most vision-based approaches, including CNNs, is the need for a considerable amount of data to achieve satisfactory accuracy. Consequently, alternatives such as GPU processing are required due to the increased computational demands.

We are also considering potential avenues for future research, including a comparative performance analysis among SVM, random forest, and CNN methodologies for detecting surface damage in geotechnical structures. Additionally, we propose exploring popular CNN architectures, facilitated by GPU acceleration and parallel processing techniques.

Furthermore, we recommend developing CNNs multiclass for detecting other types of surface failure, including seepage, leakage, sand boils, and abnormal vegetation. Moreover, it combines images and sensors to compile a dataset containing more images featuring surface damage indicators. We may enhance detection accuracy by employing a multiclass classifier for surface damage classification to address issues.

Lastly, we highlight that the ability of the model to distinguish between the 'damaged' and 'intact' classes is outstanding. It enables precise and enhanced identification of failure indicators. Early detection of failure indicators on the surface of slopes can facilitate proper maintenance and trigger alarms, helping to prevent disasters. This is crucial as the integrity of the soil directly impacts the structures constructed both around and above it. Furthermore, the integration of UAV-CNN and remote inspection can effectively minimise the need for physical on-site inspections, while simultaneously enhancing the overall safety of professionals tasked with monitoring remote and hostile areas.

### Funding

This work was supported by the Coordination for the Improvement of Higher Education Personnel (CAPES), the National Council for Scientific and Technological Development

(CNPq), Norte Energia S.A. (Code number ANEEL PD-07427-0321/2021), the Support Program for Qualified Production (PROPESP/UFPA-PAPQ) (notice 02/2023), and Institutional Program for Research Promotion of PROPESQ-CEFET-MG (the Program for Qualitative Improvement of Scientific Production–(PROMEQ-CEFET-MG)). The funders had no role in study design, data collection and analysis, decision to publish, or preparation of the manuscript.

### Grant Disclosures

The following grant information was disclosed by the authors:
The Coordination for the Improvement of Higher Education Personnel (CAPES).
The National Council for Scientific and Technological Development (CNPq).
Norte Energia S.A. (Code number ANEEL PD-07427-0321/2021), the Support Program for Qualified Production (PROPESP/UFPA-PAPQ) (notice 02/2023).
Institutional Program for Research Promotion of PROPESQ-CEFET-MG (the Program for Qualitative Improvement of Scientific Production–(PROMEQ-CEFET-MG)).

### Competing Interests

The authors declare there are no competing interests.

### Author Contributions

- Thabatta Moreira Alves de Araujo conceived and designed the experiments, performed the experiments, analyzed the data, performed the computation work, prepared figures and/or tables, authored or reviewed drafts of the article, and approved the final draft.
- Carlos André de Mattos Teixeira conceived and designed the experiments, performed the experiments, analyzed the data, performed the computation work, prepared figures and/or tables, authored or reviewed drafts of the article, and approved the final draft.
- Carlos Renato Lisboa Francês conceived and designed the experiments, performed the experiments, analyzed the data, performed the computation work, prepared figures and/or tables, authored or reviewed drafts of the article, and approved the final draft.

### Data Availability

The data is available at figshare: Araujo, Thabatta; DE ARAUJO, THABATTA (2023). Superficial_geotecnical_damage_. figshare. Figure. https://doi.org/10.6084/m9.figshare.23991339.
The code is available at figshare: Araujo, Thabatta (2023). Code_damage_detection_CNN. figshare. Software. https://doi.org/10.6084/m9.figshare.24247171.

### Supplemental Information

Supplemental information for this article can be found online at http://dx.doi.org/10.7717/peerj-cs.2052#supplemental-information.

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
