# Peer review of "Enhancing geotechnical damage detection with deep learning: a convolutional neural network approach"

_PeerJ Computer Science, doi:10.7717/peerj-cs.2052_

## Round 0.1 · original submission · Major Revisions

Based on the comments and recommendations from the reviewers, the paper needs essential revision and improvement.

Reviewer 1 ·

Basic reporting

no comment

Experimental design

no comment

Validity of the findings

The method uses a customized CNN-based architecture with five convolutional-pooling layers and two fully connected layers to classify the damaged or intact image. How about the performance of the tranditional method ,like SVM or Random Forest?How about the deeper CNN layers?

·

Basic reporting

The authors have taken pains to describe the existing literature on which they have built their binary classifier. However, there are gaps in terms of the raw data being shared, crucially the code used has not been provided in the supplementary either. Raw data should have also been made available on Figshare or some other dataset sharing platform. As written, it is not self contained, and there seems to be an over-reliance on existing research, for example, even the ADAM optimizer and its equations are omitted.

Experimental design

There are no standard training / test loss plots to judge over-fitting from. There are no baseline models nor is there enough context on if a binary classifier is useful to geotechnical damage and detection, especially since the models have no predictive ability in the wider context of geotechnical damage and detection. To "detect threats" is mentioned as the goal, but a binary classifier does not provide much in the way of preventing accidents. To improve this, consider linking this to a phenomological model of damage and trying to interpret the probabilities as degrees of damage.

Validity of the findings

As discussed above, the data is not provided, nor is the code, and the results / benefits are unclear. Additionally, the models do not have any preprocessing for time / weather / photo conditions etc.

Additional comments

Unfortunately at this level of rigor it is not possible to amend this work without a full rewrite, but I am convinced with effort, this will be useful and (most importantly) correct and reproducible research paper.

---

## Round 0.2 · Major Revisions

The recommendations from the two reviewers are divergent. Please revise the paper following all the comments from the reviewers.

Reviewer 1 ·

Basic reporting

The introduction should be compared with other Geotechnical Damage Detection methods based on deep learning. The novelty of the paper lies solely in performing a binary classification task using a 3-layer CNN. Why not employ established CNN models such as VGG, ResNet, and others?

The experimental section should include experiments comparing the proposed method with some mainstream object detection approaches.

The experimental section should include the experimental results of author's method on publicly available datasets relevant to Geotechnical Damage Detection, rather than conducting experiments solely on author's dataset.

The dataset for the paper comprises only 674 images, which is insufficient to simulate real-world scenarios or obtain meaningful experimental data.

Experimental design

The experimental section provides results in terms of a confusion matrix and ROC curve. The experimental part should include comparative tests between author's method and other object detection approaches. Additionally, applying author's method to experiments on other publicly available datasets is crucial to thoroughly demonstrate its effectiveness.

Validity of the findings

This paper introduces a simple neural network model for Geotechnical Damage Detection and proposes a dataset. However, there is insufficient evidence in the paper to validate the effectiveness of both the model and the dataset. Additionally, the current workload in the paper is insufficient to support the arguments they put forward.

·

Basic reporting

The paper has been rewritten almost entirely, which has improved the flow considerably. The authors have made a commendable effort to share data and code, the paper is now also reasonably self contained. The title change was very timely.

Experimental design

The design has not changed, but it has been better motivated. I believe there is sufficient effort here.

Validity of the findings

The results of the network described are good, and they seem robust to a certain amount of weather "noise". However, I remain unconvinced about the model's performance on images under different weather conditions, e.g. in a practical situation, one would expect pouring rain to accompany a mudslide. This is almost the equivalent of adding noise to the images, and neural networks are notorious sensitive to noise. I would suggest rewriting the conclusions. Although the results are promising from a computer science perspective, the applicability in "live" applied situations is not wholly discussed and should be rephrased.

Additional comments

I am very impressed by the improvement made. It is commendable.

---

## Round 0.3 · accepted · Accept

The paper is ready to be accepted.

·

Basic reporting

No comment.

Experimental design

No comment.

Validity of the findings

No comment.

Additional comments

I am happy to recommend acceptance, having seen this evolve into a well contained research article with clear future plans. Very commendable.